# The Critical Number and Size of Precipitation Embryos to Accelerate Warm Rain Initiation

Jung-Sub Lim[1, 2], Yign Noh[2], Hyunho Lee[3], and Fabian Hoffmann[1]

[1]Meteorologisches Institut, Ludwig-Maximilians-Universität, München, Germany.
[2]Department of Atmospheric Sciences, Yonsei University, Seoul, Republic of Korea.
[3]Department of Atmospheric Science, Kongju National University, Gongju-si, Republic of Korea.

**Correspondence:** Yign Noh (noh@yonsei.ac.kr) and Fabian Hoffmann (Fa.hoffmann@lmu.de)

**Abstract.** Understanding warm rain initiation through droplet collision and coalescence is a fundamental yet complex challenge in cloud microphysics. Although it is well-known that sufficiently large droplets, so-called precipitation embryos (PEs), may accelerate droplet collisions, it is uncertain how many and how large these PEs should be to affect rain initiation substantially. We address this question using an ensemble of box simulations with Lagrangian cloud microphysics. We find that warm rain initiation is substantially accelerated only if the PE size or number (or the product of those) exceeds a critical threshold necessary to compensate for the PE-induced suppression of collisions among non-PEs. The sensitivity of this threshold to the shape of the droplet size distribution and turbulence effects on the collision process is analyzed. It is shown that more and larger PEs are needed to accelerate rain initiation when collisions are already efficient without PEs, e.g., due to a broad droplet size distribution or strong turbulence effect. Beyond increasing our fundamental understanding of the precipitation process in warm clouds, our results may help to constrain the effect of PE-like particles intentionally or unintentionally added in climate intervention approaches, such as rain enhancement or marine cloud brightening.

## 1 Introduction

A key challenge in understanding warm rain initiation is explaining the growth of cloud droplets in the radius range between 15 and $40\,\mu m$, the so-called size gap, in which neither condensational nor collisional growth is effective (e.g., Shaw, 2003; Devenish et al., 2012; Grabowski and Wang, 2013). In the droplet size distributions (DSDs) that are too narrow or consist of too small droplets, collisions among droplets and thus precipitation formation are inefficient. These collision-limited DSDs can be regarded as being in a *collisionally stable* state (Squires, 1958), where mechanisms that accelerate the collision-coalescence process to form raindrops and initiate precipitation are crucial for breaking this stability. Research over the past five decades has identified several key mechanisms: (i) DSD broadening by entrainment and mixing (Baker et al., 1980; Blyth, 1993; Krueger et al., 1997; Lasher-Trapp et al., 2005; Cooper et al., 2013; Hoffmann et al., 2019; Lim and Hoffmann, 2023, 2024), (ii) turbulence-induced collision enhancement (TICE), which increases the collision efficiency and reduces the size dependency of droplets to initiate collisions (e.g., Saffman and Turner, 1956; Kostinski and Shaw, 2005; Pinsky et al., 2008; Wang and Grabowski, 2009; Grabowski and Wang, 2013; Onishi et al., 2015; Hoffmann et al., 2017; Chen et al., 2020; Chandrakar et al., 2024), and (iii) the role of so-called precipitation embryos (PEs) (e.g., Johnson, 1993), the primary focus of this study.

The presence of PEs larger than 20 μm can initiate the collision process as they are already larger than the size-gap range (e.g., Woodcock, 1953; Telford, 1955; Johnson, 1982; Exton et al., 1986; Johnson, 1993; Feingold et al., 1999; Teller and Levin, 2006; Alfonso et al., 2013; Hoffmann et al., 2017; Dziekan et al., 2021). The sources of these PEs can be giant aerosol particles, predominantly large sea-salt aerosols that form solution droplets having a radius between 1 μm and 100 μm (Johnson, 1982; Blyth, 1993; O'Dowd et al., 1997; Feingold et al., 1999; Jensen and Nugent, 2017; Hudson and Noble, 2020; Hoffmann and Feingold, 2023), rare (one in a million) "lucky droplets" that grow faster than the average droplet (Telford, 1955; Kostinski and Shaw, 2005; Wilkinson, 2016; Alfonso and Raga, 2017; Alfonso et al., 2019), or particles from cloud seeding experiments to enhance precipitation (Bowen, 1952; Cotton, 1982). In this study, PEs are broadly defined as large droplets, irrespective of their origin.

Although the aforementioned studies generally agree that PEs can accelerate warm rain initiation, it is uncertain how their number and size affect the acceleration of droplet growth. Some studies suggest that $10^{-3}$ cm$^{-3}$ 20 μm-sized droplets can effectively accelerate the rain initiation (e.g., Feingold et al., 1999) and change the amount of precipitation and cloud properties such as the droplet number concentration and liquid water content (e.g., Yin et al., 2000). Other studies indicate that the effectiveness of PEs relies on the type of the cloud, with shallower clouds being more susceptible (e.g., Kuba and Murakami, 2010; Dziekan et al., 2021). In the absence of PEs, DSDs with small-sized droplets barely initiate precipitation unless stochastic fluctuations in the collision process are considered. This phenomenon is known as the "lucky droplet" effect, which may produce PEs on its own (Telford, 1955; Kostinski and Shaw, 2005; Dziekan and Pawlowska, 2017). When this effect dominates, adding only a few PEs may not substantially accelerate rain initiation. In addition, although a few previous studies have investigated these mechanisms (Hoffmann et al., 2017; Chen et al., 2020), it remains unclear whether PE and TICE compete or complement each other in influencing collisional growth.

Lastly, there is a large uncertainty in the number concentration of PEs in clouds (Khain, 2009). For instance, PEs originating from 1–20 μm sea salt aerosols exhibit a wide range of concentrations from $10^{-4}$ to $10^{-2}$ cm$^{-3}$ (Jung et al., 2015; Jensen and Nugent, 2017), with a strong environmental and spatial dependency (Woodcock, 1953; Jung et al., 2015). Based on the "one in a million" definition of "lucky droplet" acting as PEs (e.g., Kostinski and Shaw, 2005), typical cloud droplet concentrations over the ocean and continents ($10^1$ to $10^3$ cm$^{-3}$) imply PE concentrations of $10^{-5}$ to $10^{-3}$ cm$^{-3}$. On the other hand, for climate-engineering practices such as cloud seeding, the concentration of seeded particles can exceed natural values, ranging from $10^{-1}$ to $10^1$ cm$^{-3}$ (Kuba and Murakami, 2010). Due to this large variability, assessing the PE effect for a broad range of PE concentrations is important.

A particle-based Lagrangian cloud model (LCM) is the natural choice for such investigation (e.g., Gillespie, 1972; Shima et al., 2009; Hoffmann et al., 2017; Dziekan and Pawlowska, 2017; Unterstrasser et al., 2020; Li et al., 2022). Particularly, it was shown that a "one-to-one" LCM, where each computational particle represents one single cloud drop is suitable to consider stochastic fluctuations in collisional growth naturally (e.g., Dziekan and Pawlowska, 2017; Li et al., 2022). While considering the numerous processes that also affect warm rain initiation (i.e., aerosol activation and condensation) is essential for investigating rain initiation, a simple box model of the collision-coalescence process alone offers unique insights that cannot be captured in a more complex model due to its tremendous computational costs when using the one-to-one LCM. Therefore,

this study aims to investigate the early stages of collisional growth to determine the number and size of PEs needed to accelerate collisional growth.

This paper is organized as follows. Section 2 introduces the LCM box model and the simulation setup. Section 3 presents the results revealing the threshold on the minimum number and size of PEs to accelerate droplet collisions. Section 4 explores the mechanism behind the existence of this threshold. We conclude our paper in Section 5.

## 2   Model and Simulation Setup

### 2.1   Lagrangian Cloud Box Model

In most applications, each computational particle of an LCM represents a large number of real droplets with identical properties, frequently called superdroplets, by introducing a weighting factor ($W_i$) (e.g., Shima et al., 2009). Thus, the number concentration of droplets is determined by

$$N = \sum_{i=1}^{n_{\mathrm{ptcl}}} \frac{W_i}{\Delta V}, \tag{1}$$

where $\Delta V$ is a reference volume, and $n_{\mathrm{ptcl}}$ represents the number of computational particles in $\Delta V$. In this study, we apply the "one-to-one" method, where each computational particle represents a single cloud droplet ($W_i = 1$). This approach fully captures the inherent stochasticity of the collision process (Shima et al., 2009; Dziekan and Pawlowska, 2017; Li et al., 2022).

The collision scheme follows the approach introduced by Shima et al. (2009) and Sölch and Kärcher (2010), in which a collision occurs with the probability

$$p_{m,n} = \frac{K_{m,n}}{\Delta V} \delta t, \tag{2}$$

primarily determined by the gravitational collection kernel

$$K_{m,n} = \pi (r_{\mathrm{m}} + r_{\mathrm{n}})^2 E(r_{\mathrm{m}}, r_{\mathrm{n}}) |w(r_{\mathrm{m}}) - w(r_{\mathrm{n}})|, \tag{3}$$

where $r_{\mathrm{m}}$ and $r_{\mathrm{n}}$ are the radii of the interacting droplets, $E$ the collision efficiency of droplet pairs (Hall, 1980), and $w$ the droplet terminal velocity (Beard, 1976). $\delta t$ is the model time step. Here, we assume the coalescence efficiency to be unity. In this study, a collected droplet is removed from the simulation after the collision-coalescence event, and the mass of the collecting droplet increases by the mass of the collected droplet.

The simulations do not consider other processes besides collisional growth, such as condensation or sedimentation, which are beyond the focus of our study. Therefore, our results should be regarded as representative for the early stages of collisional growth only. For a detailed explanation of the LCM collision scheme, readers are referred to Hoffmann et al. (2017), Noh et al. (2018), and Unterstrasser et al. (2020).

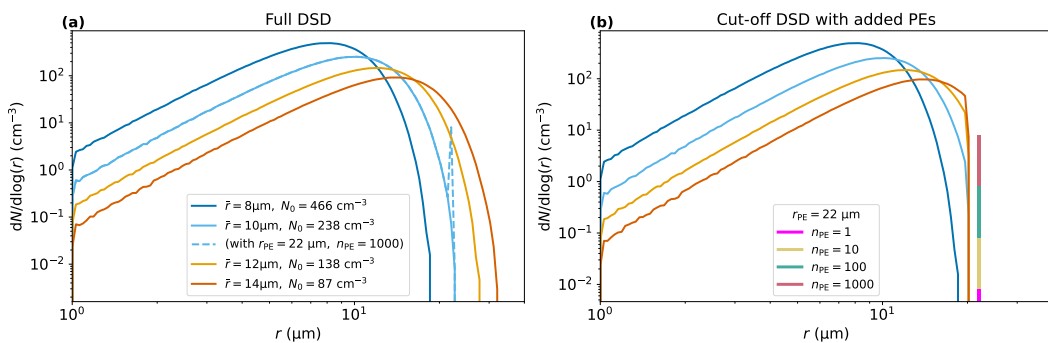

**Figure 1.** (a) Initial DSDs for various $\bar{r}$ and their corresponding $N_0$ values. The dashed line represents the DSD with $r_{\mathrm{PE}} = 27$ µm and $n_{\mathrm{PE}} = 1000$. b) Initial DSDs with a DSD cut-off for various $\bar{r}$ and $N_0$ values, along with a vertical bar plot showing various PE distributions for $r_{\mathrm{PE}} = 27$ µm and various $n_{\mathrm{PE}}$ values.

## 2.2 Simulation Setups

The initial DSD is expressed as

$$N(m) = \frac{N_0}{\bar{m}} \exp\left(-\frac{m}{\bar{m}}\right), \tag{4}$$

where $m$ is the mass of a droplet, $N_0 = 238$ cm$^{-3}$ the initial droplet number concentration, and $\bar{m}$ the mass of a droplet with $\bar{r} = 10$ µm (see light blue line in Fig. 1). The DSD results in a cloud water mixing ratio ($q_c$) of approximately $1.0$ g kg$^{-1}$. Additionally, cases with $\bar{r} = 8$, $12$, or $14$ µm are considered to investigate the effect of PEs in different DSD shapes. In these cases, $N_0 = 466$, $138$, and $87$ cm$^{-3}$ to achieve the same $q_c = 1.0$ g kg$^{-1}$ (Fig. 1). We name these cases "RM", where RM stands for the mean radius with the subsequent number denoting $\bar{r}$ (e.g., RM10). In this study, we primarily discuss the

simulation with $\bar{r} = 10$ µm, i.e., RM10, unless otherwise noted.

To explore the impact of PEs, we investigate 42 ensemble simulations, each representing different combinations of PE radii ($r_{\mathrm{PE}} = 18, 22, 27, 33, 40$, and $50$ µm) and numbers ($n_{\mathrm{PE}} = 1, 3, 10, 30, 100, 300$, and $1000$). Here, we define PEs as any droplets added to the original DSD, although the conventional definition of PEs requires $r_{\mathrm{PE}} > 20$ µm. The largest PE size is chosen to correspond to the size of haze particles grown from $1-5$ µm sea-salt aerosols (Kuba and Murakami, 2010). We choose a

100 minimum $n_{\mathrm{PE}} = 1$ to investigate whether a 'one in a million' PE can accelerate droplet collision, as highlighted in previous studies on lucky droplets (Kostinski and Shaw, 2005; Dziekan and Pawlowska, 2017). Within a given reference volume, the minimum and maximum $n_{\mathrm{PE}}$ values of 1 and 1000 correspond to concentrations of approximately $2.97 \times 10^{-4}$ cm$^{-3}$ and $2.97 \times 10^{-1}$ cm$^{-3}$, respectively, reflecting the wide range of PE concentrations observed in nature (Khain, 2009; Jung et al., 2015).

Every setup is simulated 100 times with different random numbers to ensure statistical convergence (cf. Fig. A1). Using a timestep $\delta t = 0.1$ s, the model is integrated for 7200 s to account for the slowest realization to complete collisional growth, but the discussion is focused on the initial 2500 s, capturing the initiation of collisional growth. A total of $10^6$ computational

particles ($n_{\text{ptcl}} = 10^6$) are initialized to represent the initial DSD of RM10, resulting in a reference volume $\Delta V = 3.36 \times 10^{-3}$ m$^3$. For cases with different $N_0$, $n_{\text{ptcl}}$ scales with $N_0$ from $10^6$ at RM10 ($N_0 = 238$ cm$^{-3}$) to $n_{\text{ptcl}} = 1,953,125$ for RM8 ($N_0 = 466$ cm$^{-3}$), and $n_{\text{ptcl}} = 364,431$ for RM14 ($N_0 = 87$ cm$^{-3}$). This adjustment applies only to non-PE particles, with $n_{\text{PE}}$ being varied from 1 to 1000 for all $N_0$.

Changing $\bar{m}$ also alters the number and mean radius of droplets larger than 20 μm, which are critical for initiating droplet collisions. For instance, the radii of the largest initialized droplets are 24 and 34 μm for RM10 and RM14, respectively. To isolate the dependency of the PE effect on the DSD shape for smaller droplets, we remove droplets larger than 20 μm in specific simulations (Wang et al., 2006; Dziekan and Pawlowska, 2017). This initialization is referred to as a "cut-off DSD" (see the dotted line in Fig. 1). We denote these cases by adding the letter "N" to the naming convention (e.g., RM10N), referring to the resulting narrower DSD.

To investigate the effect of TICE, five different kinetic energy dissipation rates $\varepsilon = 5$, 10, 50, 100, and 200 cm$^2$ s$^{-3}$ are considered for RM10. These $\varepsilon$ values are chosen to explore the TICE effect across different cloud types, where typical values range from $1 - 10$ cm$^2$ s$^{-3}$ in stratocumulus clouds, $10 - 100$ cm$^2$ s$^{-3}$ in shallow convective clouds, and $100 - 1000$ cm$^2$ s$^{-3}$ in deep convective clouds (Siebert et al., 2006; Seifert et al., 2010; Pruppacher and Klett, 2012). TICE is incorporated in Eq. 3 using the parameterizations developed by Ayala et al. (2008) and Wang and Grabowski (2009), which are steered by $\varepsilon$. When TICE is considered, the case names are amended by a T followed by the numerical value of $\varepsilon$ in cm$^2$ s$^{-3}$(e.g., RM10-T100).

In this study, the timescales $t_{100}$ and $t_{10\%}$ are used to characterize the precipitation efficiency. In previous studies, the time for the first raindrop formation is used to quantify the efficiency of stochastic raindrop formation (Dziekan and Pawlowska, 2017). In this study, a raindrop is defined as a droplet larger than 40 μm. As PEs considered in this study can be raindrops already, we define $t_{100}$ as the time required for the formation of the first 100 μm droplet, i.e., a sufficiently large droplet that stimulates subsequent collisions (Kostinski and Shaw, 2005; Alfonso et al., 2019). Thus, $t_{100}$ characterizes the efficiency for *raindrop formation*. The timescale $t_{10\%}$ represents the time when 10 % of the initial cloud droplet mass converts to rain, measuring the efficiency of *rain initiation* from a mass perspective (Onishi et al., 2015; Dziekan and Pawlowska, 2017).

Adding PEs increases the initial $q_{\text{c}}$, or the rainwater mixing ratio $q_{\text{r}}$ when $r_{\text{PE}} > 40$ $\mu$m and $n_{\text{PE}} > 0$, potentially limiting the comparability of simulated cases. To address this, we restricted the analysis of $t_{10\%}$ and further conversion rates such as autoconversion rate (i.e., raindrop formation by collisions between cloud droplets), and accretion rate (i.e., raindrop growth by raindrops collecting cloud droplets) to cases where the increase in the initial $q_{\text{c}} + q_{\text{r}}$ due to the addition of PEs is below 2 %. In most cases, the increase in $q_{\text{c}}$ and $q_{\text{r}}$ is below 1 %. However, two exceptions, $n_{\text{PE}} = 300$, with $r_{\text{PE}} = 40$ μm and $n_{\text{PE}} = 1000$, with $r_{\text{PE}} = 27$ μm, show an increase of 1.9 %.

## 3   PE Effect on Precipitation Timescales

### 3.1   Critical Thresholds for Raindrop Formation and Rain Initiation

Figure 2 shows the ensemble-averaged $t_{100}$ and $t_{10\%}$, named $\mu_{100}$ and $\mu_{10\%}$, for RM10 and RM14. In general, increasing $r_{\text{PE}}$ and $n_{\text{PE}}$ both shorten $\mu_{100}$ and $\mu_{10\%}$, indicating accelerated rain initiation. However, when $r_{\text{PE}} = 18$ μm, i.e., smaller than

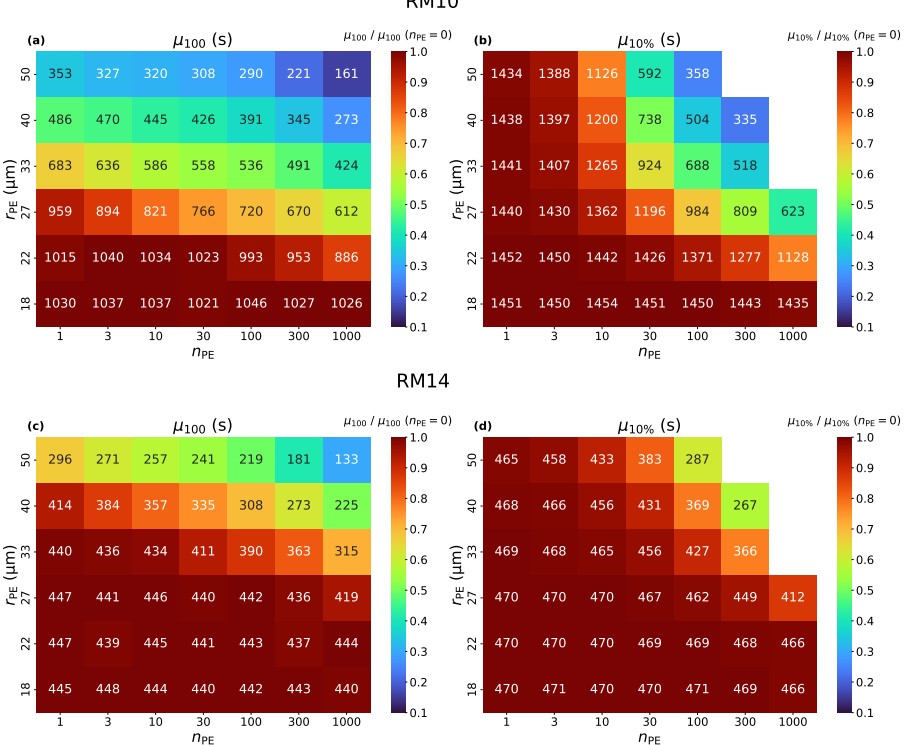

**Figure 2.** Ensemble-averaged values of (a and c) time for the first 100 μm raindrop formation, $\mu_{100}$, and (b and d) time for 10 % of cloud droplets to convert to raindrops, $\mu_{10\%}$, for RM10 (first row) and RM14 (second row). The abscissa represents $n_{\mathrm{PE}}$, the ordinate $r_{\mathrm{PE}}$. $\mu_{10\%}$ values for the cases where the initial $q_c + q_r$ increase due to PEs by more than 2 % are not shown. Colors in the plot represent the ratio of $\mu_{100}$ and $\mu_{10\%}$ to their values in the case without PEs ($n_{\mathrm{PE}} = 0$). In the case without PEs, $\mu_{100} = 1027$ s and $\mu_{10\%} = 1452$ s for RM10, and $\mu_{100} = 442$ s and $\mu_{10\%} = 470$ s for RM14.

the maximum droplet radius of the initial DSD (Fig. 1a), $\mu_{100}$ and $\mu_{10\%}$ are not substantially accelerated compared to those cases without PEs regardless of $n_{\mathrm{PE}}$. Note that, in the case without PEs, $\mu_{100}$ and $\mu_{10\%}$ are 1027 s and 1452 s, respectively, for RM10. This indicates that the addition of PEs smaller than the maximum droplet radius of the DSD, even in large numbers (e.g., $n_{\mathrm{PE}} = 1000$), has a negligible effect on raindrop formation. Interestingly, $n_{\mathrm{PE}}$ plays a more crucial role for $\mu_{10\%}$ than for $\mu_{100}$. For $n_{\mathrm{PE}} \leq 3$, $\mu_{10\%}$ is not accelerated (Fig. 2b) even for large PEs, whereas $\mu_{100}$ is accelerated (Fig. 2a). Thus, a faster $\mu_{100}$ does not always ensure a shorter $\mu_{10\%}$.

For RM10, when $n_{\mathrm{PE}} = 3$, the PE number concentration is approximately $10^{-3}$ cm$^{-3}$. In this case, even PEs larger than 40 μm are not effective in accelerating $t_{10\%}$ (Fig. 2b). However, when the PE concentration increases to a relatively high value ($n_{\mathrm{PE}} = 30$), PEs larger than 22 μm can substantially accelerate $t_{10\%}$ (Fig. 2b). Such high PE concentrations are uncommon but have been observed in certain oceanic conditions (Jung et al., 2015). In contrast, for RM14, which represents typical maritime clouds in a pristine environment with $N_0 = 87$ cm$^{-3}$, the effect of PEs is reduced. PEs smaller than 33 μm are unable

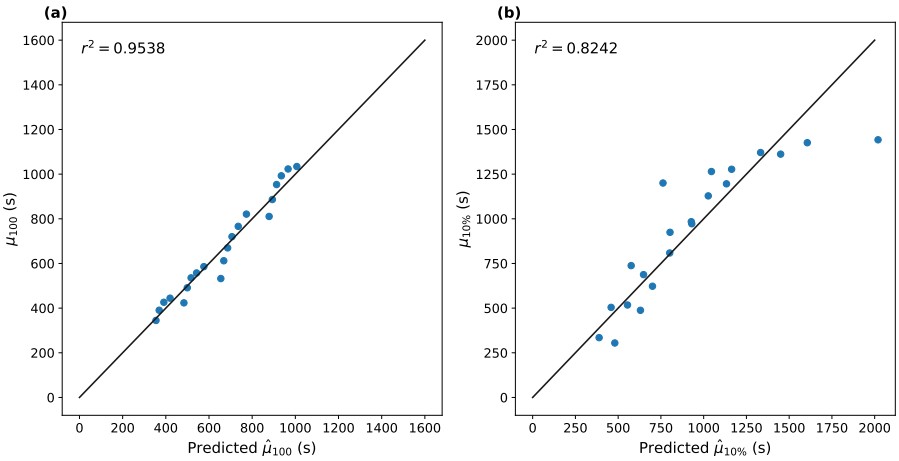

**Figure 3.** Scatter plots of simulated (a) $\mu_{100}$ and (b) $\mu_{10\%}$ (ordinate) and predicted values (abscissa) using Eq. (5) for RM10. The solid black lines indicate the one-to-one line.

to accelerate $\mu_{100}$ regardless of $n_{\mathrm{PE}}$ (Fig. 2c). Moreover, $t_{10\%}$ is accelerated only when both $n_{\mathrm{PE}}$ and $r_{\mathrm{PE}}$ are very large (Fig. 2d). However, such extreme conditions are uncommon in typical maritime environments. This suggests that the impact of PEs depends on the initial DSD shape, requiring a collisionally stable cloud for a substantial effect.

Overall, Fig. 2 shows $\mu_{100}$ and $\mu_{10\%}$ can be shortened with increasing $n_{\mathrm{PE}}$ and $r_{\mathrm{PE}}$, but only if a critical threshold is exceeded. Below this critical threshold, the effect of PEs on rain initiation is negligible. This raises the following question: What are the specific size and number of PEs required to accelerate rain initiation substantially? To identify the critical threshold, we first express $\mu_{100}$ and $\mu_{10\%}$ as functions of $n_{\mathrm{PE}}$ and $r_{\mathrm{PE}}$. As shown in Fig. 2, $\mu_{100}$ and $\mu_{10\%}$ decrease as both $n_{\mathrm{PE}}$ and $r_{\mathrm{PE}}$ increase once the critical threshold is exceeded. Thus, we write

$$\mu_\alpha = c_\alpha - k_\alpha n_{\mathrm{PE}}^{a_\alpha} r_{\mathrm{PE}}^{b_\alpha} = c_\alpha - k_\alpha \Phi_\alpha(n_{\mathrm{PE}}, r_{\mathrm{PE}}) \tag{5}$$

for a $\mu_\alpha$ exceeding the critical threshold. Here $k_\alpha$ is a rate-of-change coefficient, $c_\alpha$ a constant, and $\Phi_\alpha(n_{\mathrm{PE}}, r_{\mathrm{PE}})$ represents the composite relationship of $n_{\mathrm{PE}}$ and $r_{\mathrm{PE}}$ with scaling exponets $a_\alpha$ and $b_\alpha$. The subscript $\alpha$ is 100 or 10% for $\mu_{100}$ and $\mu_{10\%}$, respectively.

     To determine the parameters of Eq. (5), we fit $a_\alpha$, $b_\alpha$, $c_\alpha$, and $k_\alpha$, using $\mu_{100}$ and $\mu_{10\%}$ from cases with $r_{\mathrm{PE}} \geq 22$ and

$n_{\mathrm{PE}} \geq 10$. In these cases, both $\mu_{100}$ and $\mu_{10\%}$ are directly affected by changes in $r_{\mathrm{PE}}$ and $n_{\mathrm{PE}}$ (Fig. 2), i.e., the PE critical threshold is exceeded. The fitted parameters are $a_{100} = 0.086$, $b_{100} = 3.086$, $c_{100} = 3363$ s, and $k_{100} = 1.035$ for $\mu_{100}$, and $a_{10\%} = 0.13$, $b_{10\%} = 1.13$, $c_{10\%} = 165592$ s, and $k_{10\%} = 3.018$ for $\mu_{10\%}$ with $r_{\mathrm{PE}}$ in μm. Units of each parameter are detailed in Appendix B.

     Our focus will be on $\Phi_\alpha(n_{\mathrm{PE}}, r_{\mathrm{PE}})$ with $a_\alpha$ and $b_\alpha$ first. The parameters $c_\alpha$ and $k_\alpha$ will be discussed in more detail after

we expand Eq. (5) with more physically meaningful terms. The values of $a_\alpha$ and $b_\alpha$ indicate that both $\mu_{100}$ and $\mu_{10\%}$ are more sensitive to $r_{\mathrm{PE}}$ than $n_{\mathrm{PE}}$ as expected from Fig. 2. When comparing $a_{100}$ and $b_{100}$ to $a_{10\%}$ and $b_{10\%}$, the dependency on $n_{\mathrm{PE}}$

is stronger in $\mu_{10\%}$ than in $\mu_{100}$, which is also consistent with the results shown in Fig. 2. Figure 3 juxtaposes the simulated and predicted $\mu_{100}$ and $\mu_{10\%}$ using Eq. (5). This result indicates that $\mu_{100}$ and $\mu_{10\%}$ can be expressed with $\Phi_\alpha$ relatively well. However, Eq. (5) overestimates $\mu_{10\%}$ when it is over 1400 s (Fig. 3b). This is due to the cases with $n_{PE} < 10$, which show almost no dependency on $r_{PE}$. The reasons behind this behavior will be further discussed in detail in Sec. 4.

To better capture the behavior of $\mu_{100}$ and $\mu_{10\%}$, especially near the critical threshold where the dependency on $r_{PE}$ and $n_{PE}$ vanishes, e.g., cases where $r_{PE} < 22$ μm and $n_{PE} < 10$ for RM10, we expand Eq. (5) by a Heaviside step function $\mathcal{H}$, such that

$$\mu_\alpha(\Phi_\alpha) = \mu_{\alpha,c} - k_\alpha(\Phi_\alpha - \Phi_{\alpha,c}) \cdot \mathcal{H}(\Phi_\alpha - \Phi_{\alpha,c}), \tag{6}$$

where $\mu_{\alpha,c}$ is the baseline value of $\mu_\alpha$ in the absence of PEs incorporating parameter $c_\alpha$ from above. When fitting Eq. (6) to all results, the parameters $a_\alpha$ and $b_\alpha$ were fixed to the values previously obtained from RM10 to enable a more direct comparison between different cases, focusing solely on the parameters in Eq. (6). The specific parameters for Eq. (6) and their r-squared values are detailed in Appendix B. In general, r-squared values exceed 0.95 for $\mu_{100}$, and range from 0.67 to 0.84 for $\mu_{10\%}$. The results of $\mu_{100}(\Phi_{100})$ and $\mu_{10\%}(\Phi_{10\%})$ for RM10 are shown in Fig. 4a and b as blue solid lines. Until exceeding their critical thresholds ($\Phi_{100,c} = 1.91 \times 10^4$ and $\Phi_{10\%,c} = 7.23 \times 10^1$ for RM10; see Tabs. B1 and B2), $\mu_{100}$ and $\mu_{10\%}$ remain constant at $\mu_{100,c} = 1025$ s and $\mu_{10\%,c} = 1405$ s. These values agree well with $\mu_{100}$ and $\mu_{10\%}$ without PEs, 1027 s and 1452 s, respectively. However, once $\Phi_\alpha$ becomes larger than $\Phi_{\alpha,c}$, i.e., exceeds the critical threshold, $\mu_{100}$ and $\mu_{10\%}$ decrease as expected from Eq. (5).

### 3.2 Factors Controlling the Critical Threshold

Using Eq. (6), we are now able to investigate how the critical threshold varies for different initial DSD shapes (characterized by $\bar{r}$ and the consideration of a cut-off radius) and the presence of TICE. To achieve this, we fit the results to Eq. (6) for RM8, RM10, RM12, and RM14 without or with cut-off DSD (Fig. 4a, b, c, and d). Additionally, we consider TICE for RM10 (Fig. 4e and f). Although the values of parameters $a_\alpha$ and $b_\alpha$ for $\Phi_{100}$ and $\Phi_{10\%}$ may vary for different cases, we fix them to the values obtained earlier (see Fig. 3) to directly compare $\mu_{\alpha,c}$, $\Phi_{\alpha,c}$ and $k_\alpha$ across different initial conditions. The fitted parameters for these initial conditions are detailed in Appendix B. Figure 4 shows that all cases exhibit the same fundamental feature: the presence of a critical threshold $\Phi_{\alpha,c}$.

As $\bar{r}$ increases, both $\mu_{100,c}$ and $\mu_{10\%,c}$ decrease (Figs. 4a and b; and 5b and d). This is due to the increased number of large droplets making collisions more likely when $\bar{r}$ increases. Results from the cases with a cut-off DSD with different $\bar{r}$ are shown in Fig. 4c and d. As before, $\mu_{100,c}$ and $\mu_{10\%,c}$ also decrease with increasing $\bar{r}$ although the largest droplet size remains unchanged due to the cut-off DSD (Fig. 5b). Here, this is due to the increased number of droplets in the $15 - 20$ μm size range among non-PE droplets (Fig. 1b), which can initiate collisions through stochastic processes. However, $\mu_{100,c}$ and $\mu_{10\%,c}$ remain nearly unchanged for the $\bar{r} = 12$ μm and $\bar{r} = 14$ μm cases because the difference in the number concentration of $15 - 20$ μm droplets between these cases is minimal (cf. Fig. 1b), even though the number concentration of smaller droplets

is substantially lower for $\bar{r} = 14\mu m$. This suggests that $\mu_{100,c}$ is more sensitive to the number concentration of larger droplets (e.g., those with radii of 15–20 $\mu m$) than to smaller droplets, particularly when considering the cut-off DSD.

The critical threshold $\Phi_{100,c}$ increases with increasing $\bar{r}$, indicating that more and larger PEs are required to exceed the critical threshold (Fig. 5a). Interestingly, for $\bar{r} = 8$ $\mu m$, almost all sizes and numbers of PEs are effective in shortening $\mu_{100}$ (Fig. 2a). In contrast, for $\bar{r} = 14$ $\mu m$, $\Phi_{100,c}$ becomes very high, making most PEs ineffective in shortening $\mu_{100}$. This suggests that the PE effect is more pronounced for DSDs where collisions among droplets are less efficient, i.e., cases with smaller $\bar{r}$ and slower $\mu_{100}$.

In cases with cut-off DSD, $\Phi_{100,c}$ is lower compared to those without, for the same $\bar{r}$ (Fig. 5a). This is due to the absence of larger droplets in the initial DSD, which are as effective as PEs in the collision process. Thus, employing a cut-off DSD, which removes large droplets, amplifies the influence of PEs. Likewise, the results for $\Phi_{10\%,c}$ show a similar pattern to those for $\Phi_{100,c}$ (Fig. 5a and c), with $\Phi_{10\%,c}$ increasing as $\bar{r}$ becomes larger. Now, we examine the relationship between the radius of the largest initialized non-PE droplet and the radius of a single PE which can accelerate $t_{100}$. We employ $\Phi_{100,c}$ for RM10 and RM10N, from Table B1. When $n_{PE} = 1$, the critical $r_{PE} = 24.4$, and 22.9 $\mu m$ for RM10 and RM10N, respectively. In each case, the radius of the largest initialized non-PE droplet is 24 $\mu m$ (RM10) and 20 $\mu m$ (RM10N). Therefore, for RM10, a single PE only slightly larger than the largest initialized non-PE droplet radius is sufficient to exceed the critical threshold. In contrast, RM10N requires a PE much larger than the largest non-PE droplet. This difference implies that the critical PE size is influenced not only by the radius of the largest initialized non-PE droplet but also by the collision efficiency among non-PE droplets, which depends on $\bar{r}$ and whether a DSD cut-off is present.

Additionally, the TICE effect is considered for RM10 (Fig. 4e and f). TICE has a greater impact on $\mu_{\alpha,c}$ than on $\Phi_{\alpha,c}$. Specifically, $\mu_{\alpha,c}$ decreases as $\varepsilon$ increases (Fig. 4e, f and Fig. 5b, d), whereas $\Phi_{\alpha,c}$ exhibits only a slight increase (Fig. 4e, f and Fig. 5a, c). This indicates that TICE enhances the efficiency of collisions among all cloud droplets, making $\mu_{\alpha,c}$ shorter. Therefore, more PEs are required in the presence of TICE compared to cases without. Notably, the critical PE threshold increases substantially only when $\varepsilon \geq 200$ cm$^2$s$^{-3}$ (Fig.5a and c; Table B1). This indicates that the influence of TICE on limiting the PE effect is primarily important in deep convective clouds or in regions within shallow clouds where $\varepsilon$ is locally high (e.g., Pruppacher and Klett, 2012). In summary, when droplet collisions are already efficient without PEs—either due to the presence of large droplets (i.e., a large $\bar{r}$ or the absence of a DSD cut-off) or under the influence of TICE—a larger PE size and number are necessary to substantially accelerate rain initiation.

Although we have identified the existence of the critical threshold for the PE effect, there remains a question regarding why $t_{10\%}$ is not always affected by the presence of PEs, even though $t_{100}$ is decreased (e.g., $n_{PE} < 10$ cases in Fig. 2). This discrepancy may arise because $t_{10\%}$ involves interactions among multiple droplets and PEs, whereas $t_{100}$ depends on the behavior of an individual droplet or PE. This suggests that while PEs can accelerate the formation of the largest raindrop, these droplets may not directly impact the overall rain mass growth when the number of PEs is low. In the following section, we will explore how PEs affect $t_{10\%}$ to explain why a shorter $t_{100}$ does not ensure a shorter $t_{10\%}$.

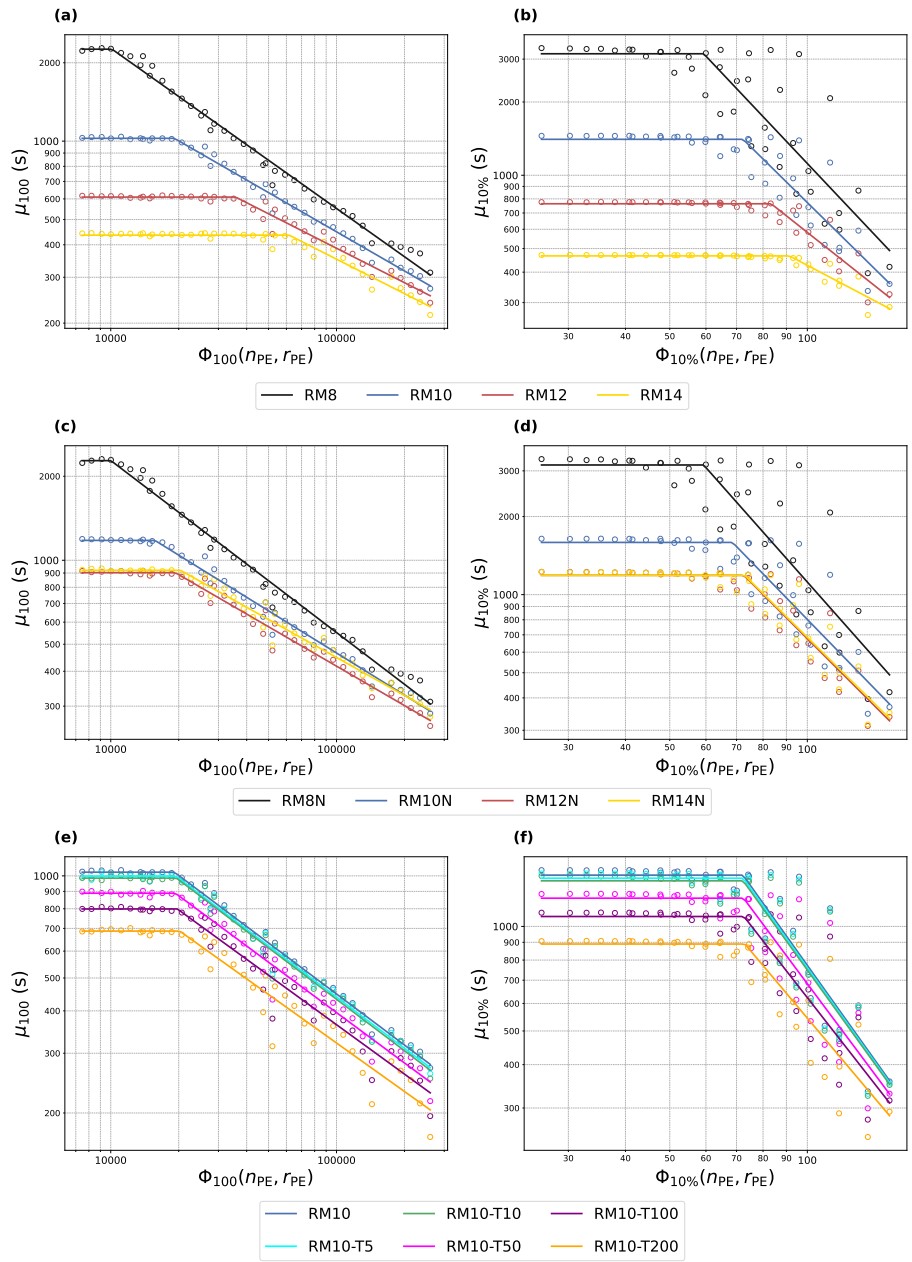

**Figure 4.** $\mu_{100}$ (left column) and $\mu_{10\%}$ (right column) as a function of $\Phi_\alpha$ for different initial conditions are shown. Each point represents the simulation results, while solid lines indicate the fitted Eq. (6). The first row (a and b) represents cases without cut-off DSD (RM8, RM10, RM12, and RM14), and the second row (c and d) represents cases with cut-off DSD (RM8N, RM10N, RM12N, and RM14N). The third row (e and f) represents RM10 with different $\varepsilon$ values (RM10-T5, RM10-T10, RM10-T50, RM10-T100, and RM10-T200).

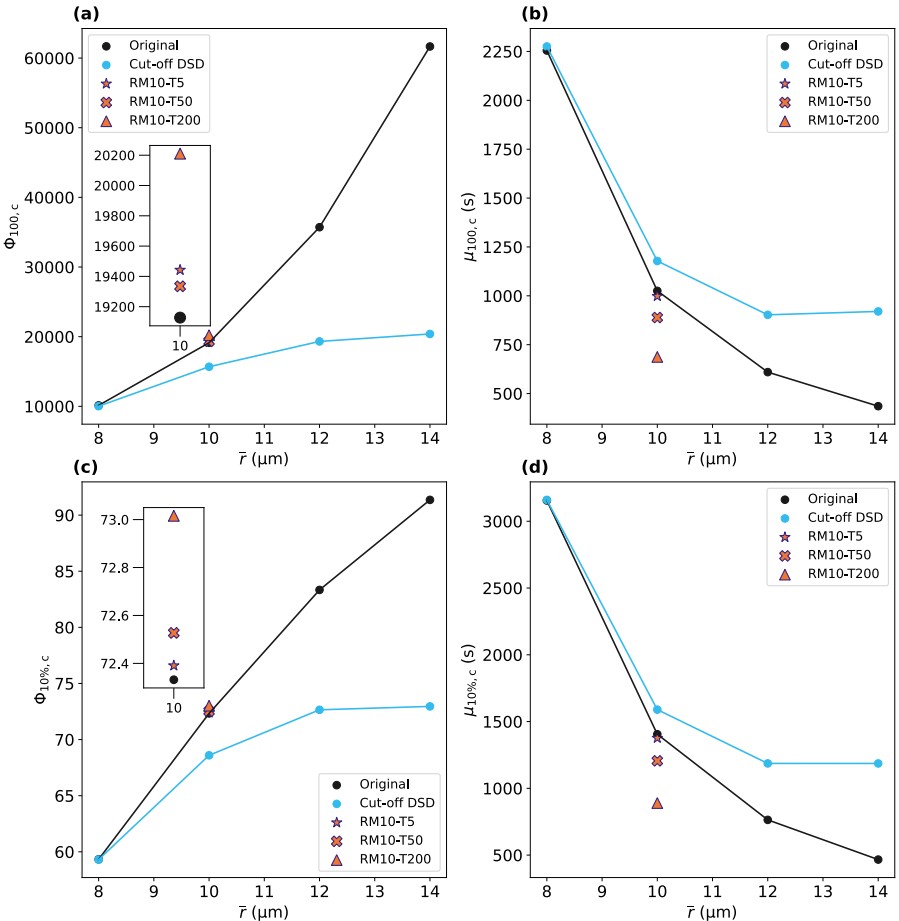

**Figure 5.** Results of (a) $\Phi_{100,c}$, (b) $\mu_{100,c}$, (c) $\Phi_{10\%,c}$ and (d) $\mu_{10\%,c}$ for different $\bar{r}$. Black circles depict cases without cut-off DSD (original), while light blue circles depict the cases with cut-off DSD. Orange star, cross, and triangle shape represent the results with $\varepsilon = 5, 50, 200 \, \mathrm{cm}^2 \, \mathrm{s}^{-3}$, respectively, for the RM10. The insets show a zoomed-in view of the cases with different $\varepsilon$ at $\bar{r} = 10 \mu m$.

## 4 PE Effects on Rain Initiation

In order to understand the effects of PE size and number on rain initiation more clearly, we consider the time series of raindrop mixing ratio $q_r$, autoconversion, and accretion rate (i.e., raindrop growth by raindrops collecting cloud droplets) using $r_{\mathrm{PE}} = 22 \, \mu m$ and $27 \, \mu m$ with different $n_{\mathrm{PE}}$ from 0 to 300 for RM10 (Fig. 6). Overall, $q_r$ evolves faster for larger $r_{\mathrm{PE}}$ and $n_{\mathrm{PE}}$ (Fig. 6a and b). However, with PEs below the critical threshold (i.e., for $n_{\mathrm{PE}} \leq 30$ at $r_{\mathrm{PE}} = 22 \, \mu m$ and $n_{\mathrm{PE}} \leq 3$ at $r_{\mathrm{PE}} = 27 \, \mu m$), the difference from the case with and without PE is insignificant, implying that PEs do not substantially enhance rain initiation, although raindrop formation ($q_r > 0$) starts earlier (Fig. 6a and b). This result is consistent with Fig. 2, in which $\mu_{100}$ is smaller than $\mu_{100,c}$, but $\mu_{10\%}$ is comparable to $\mu_{10\%,c}$.

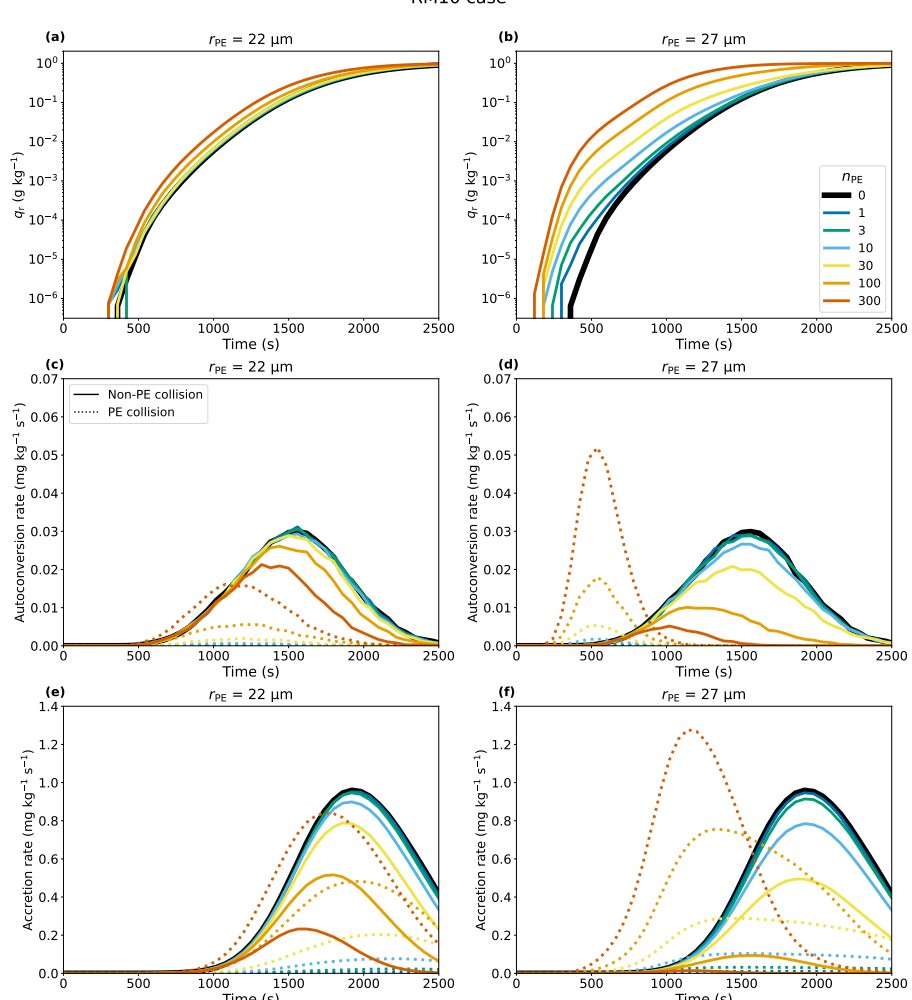

**Figure 6.** Time series of (a/b) raindrop mixing ratio, (c/d) autoconversion rate, and (e/f) accretion rate for RM10, shown for two different values of $r_{PE}$: 22 μm (first column) and 27 μm (second column). The colors of the lines represent different $n_{PE}$ values, with the black solid line representing the result from the simulation without PEs ($n_{PE} = 0$). In (c) to (f), the solid lines denote autoconversion and accretion without PEs (between non-PE droplets exclusively), while the dotted line depicts autoconversion and accretion by PEs.

The time series of autoconversion and accretion provide more details on how PEs affect rain initiation. In Fig. 6c to f, solid lines represent droplet growth without PEs (i.e., between non-PE droplets exclusively), while dotted lines represent droplet growth involving PEs (i.e., collisions between PEs and non-PE droplets or among PEs). We find that non-PE autoconversion decreases with increasing $n_{PE}$ (Fig. 6c and d). This is because large PEs have an advantage in the autoconversion process, growing faster and collecting non-PE droplets, which in turn suppresses the autoconversion of non-PE droplets.

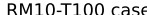

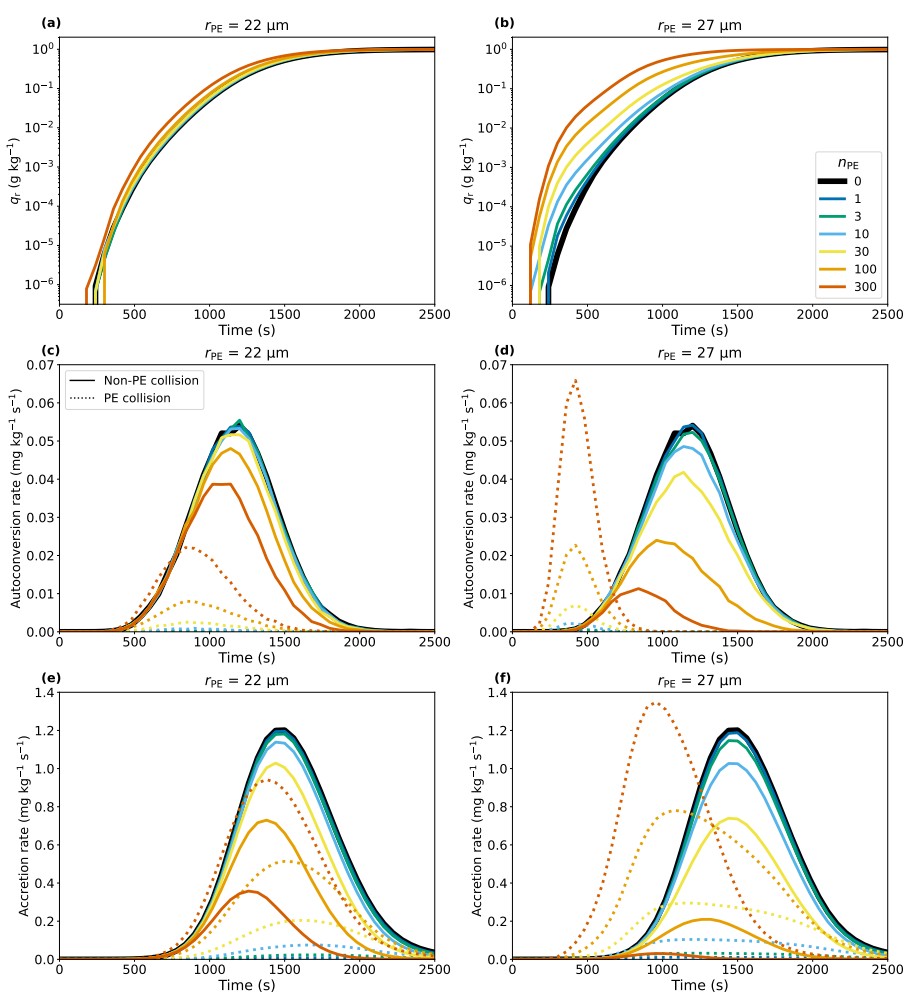

**Figure 7.** Same as for Fig. 6 but for cases with TICE using $\varepsilon = 100 \text{ cm}^2 \text{ s}^{-3}$.

For $r_{\text{PE}} = 22$ μm, both autoconversion and accretion initiate earlier with PEs than in the case without PEs, but only for $n_{\text{PE}} \geq 100$ (Fig. 6c). When $n_{\text{PE}} < 30$, autoconversion and consequently accretion by PEs are even slower than those of non-PE droplets. This implies that the collisional growth of PEs is not necessarily faster than the collisional growth between non-PE droplets. Thus, although larger PEs are more likely to collide, the overall collision frequency remains low when $n_{\text{PE}}$ is small, resulting in slower PE autoconversion compared to non-PE autoconversion. While non-PE-autoconversion always decreases with increasing $n_{\text{PE}}$, PE-autoconversion increases substantially only for $n_{\text{PE}} \geq 100$. Therefore, before exceeding the critical threshold, PEs suppress non-PE autoconversion more than they enhance autoconversion which can even lead to a decrease in the total (PE and non-PE) autoconversion. Hence, shorter $t_{100}$ does not necessarily lead to a shorter $t_{10\%}$ when $n_{\text{PE}}$ is small (Fig. 2).

For $r_\mathrm{PE} = 27$ μm, while non-PE-autoconversion always decreases with increasing $n_\mathrm{PE}$, PE-autoconversion increases substantially only when $n_\mathrm{PE} \geq 100$. Therefore, before exceeding the critical threshold, PEs suppress non-PE autoconversion more than they enhance autoconversion which can even lead to a decrease in the total (PE and non-PE) autoconversion. Interestingly, increasing $n_\mathrm{PE}$ does not affect the time to initiate PE autoconversion, but affects only its magnitude (Fig. 6c). The initiation time for PE autoconversion is influenced by $r_\mathrm{PE}$ since this process is closely related to the number of collisions or time required for droplets to grow larger than 40 μm, which occurs more quickly for larger PEs (cf. Fig. 6c and d). Thus, $r_\mathrm{PE}$ determines the initiation time for autoconversion, especially when $r_\mathrm{PE} \geq 27$ μm, while $n_\mathrm{PE}$ determines how much non-PE droplet autoconversion and accretion are suppressed. PE accretion starts earlier when $r_\mathrm{PE} = 22$ μm and $n_\mathrm{PE} > 100$ and any $n_\mathrm{PE}$ for $r_\mathrm{PE} = 27$ μm (Fig. 6e and f), which is triggered by the earlier raindrop formation by PE autoconversion (Fig. 6c and d). However, even for $r_\mathrm{PE} = 27$ μm, accretion by PEs increases only slightly when $n_\mathrm{PE} \leq 30$, i.e., below the critical threshold. Once the critical threshold is exceeded, particularly for $n_\mathrm{PE} > 30$, accretion is substantially increased and accelerated compared to the case for $n_\mathrm{PE} = 0$ (Fig. 6e and f). In this case, accretion is dominated by PEs, outweighing the decrease in non-PE autoconversion (Fig. 6e and f), and initially larger $q_\mathrm{r}$ persists (Fig. 6a and b).

Interestingly, at high $n_\mathrm{PE}$, the non-PE autoconversion and accretion rates reach their peak values earlier than in cases without PEs or with low $n_\mathrm{PE}$ (Fig. 6c, d, e, and f). During the initial 1000 s, the non-PE autoconversion rate is nearly identical across all cases, regardless of $n_\mathrm{PE}$. However, when $n_\mathrm{PE}$ is high, more non-PE droplets are collected by PEs, reducing the number of droplets available for autoconversion. As a result, the non-PE autoconversion rate peaks and declines earlier in cases with higher PE concentrations. This suppression of non-PE autoconversion decreases the number of non-PE raindrops and the non-PE accretion rate. These findings highlight that the primary role of PEs is to collect non-PE droplets, which might suppress non-PE autoconversion and accretion.

Results with TICE ($\varepsilon = 100$ cm$^2$ s$^{-3}$, Fig. 7) also highlight the importance of PEs in suppressing non-PE autoconversion. With TICE, collisions between small and similar-sized droplets are more efficient (Pinsky et al., 2008). Thus, with TICE, non-PE autoconversion is still substantial when $n_\mathrm{PE} \geq 100$ (Fig. 7d), while it is almost totally suppressed without TICE (Fig. 6d). Thus, more and larger PEs are needed to outweigh non-PE accretion, making droplet growth less sensitive to PEs when TICE is considered. However, even with TICE, if $n_\mathrm{PE}$ substantially exceeds the critical threshold ($r_\mathrm{PE} = 27$ μm and $n_\mathrm{PE} = 300$), droplet collisional growth is entirely dominated by PEs (purple solid line in Fig. 7f). Thus, while both PEs and TICE accelerate droplet collisional growth, each effect becomes weaker when the other effect dominates rain initiation (e.g., Chandrakar et al., 2024).

## 5 Summary and Conclusion

Understanding whether precipitation embryos (PEs), particles larger than the so-called size gap range, can accelerate the droplet collision process remains a key question in warm rain initiation. Despite decades of research on the effect of PEs on rain initiation (e.g., Telford, 1955; Johnson, 1982; Feingold et al., 1999; Teller and Levin, 2006; Alfonso et al., 2013), this challenge persists and is still highlighted in recent studies (e.g., Chen et al., 2020; Dziekan et al., 2021; Chandrakar et al.,

2024), underscoring the need for further investigation. In this study, we systematically investigated how PEs affect droplet collisional growth using ensembles of Lagrangian cloud model (LCM) collision simulations. Our primary focus was to identify the minimal PE size and number necessary to accelerate the droplet collision-coalescence process substantially. We evaluated the droplet collision efficiency using two timescales: the time required for the first 100 μm droplet to form ($t_{100}$) and the time to convert 10% of the total initial cloud mass to rain mass ($t_{10\%}$).

We found that the droplet collision process does not substantially accelerate when the number or size of PEs is below a critical threshold. $t_{100}$ is accelerated only when the radii of PEs are larger than the maximum non-PE droplet radius of the initial DSD. This is because $t_{100}$ is more related to the growth of a single droplet where larger droplets, such as PEs, are expected to grow faster than smaller droplets. In contrast, $t_{10\%}$ depends more on the number of PEs. Even with substantially large PEs, a faster formation of the first large raindrop does not always ensure faster rain initiation when the number of PEs is small. This is because PEs increase autoconversion and accretion only when their number is sufficient while simultaneously suppressing the autoconversion of non-PE droplets to become raindrops. Thus, when autoconversion of non-PE droplets is already efficient, more or larger PEs are required to accelerate $t_{10\%}$.

To determine the critical threshold for rain initiation by PEs, we derived a simple equation that relates the number and size of PEs to $t_{100}$ and $t_{10\%}$. The equation revealed that the critical threshold depends on the *collisional stability* of the droplet size distribution (DSD) characterized by the DSD shape or turbulence-induced collision enhancement (TICE). We showed that increasing the droplet mean radius and hence the size of pre-existing large droplets increases the *collisional stability* of the DSD and makes the collisional process less susceptible to PE perturbations because non-PE droplet collisions are already sufficient for initiating rain. Equivalently, more and larger PEs are needed to substantially accelerate the droplet growth with TICE, which increases the collision frequency among smaller non-PE droplets making the collision process less reliant on PEs. Although TICE does not directly alter the PE critical threshold, it reduces the difference in rain initiation acceleration between cases with and without PEs. Consequently, more and larger PEs are required to achieve the same acceleration in droplet growth as in cases without TICE.

In this study, PEs larger than 22 μm are found to effectively accelerate the precipitation ($t_{10\%}$) for clouds in relatively polluted environments, when their concentration exceeds $10^{-3}$ cm$^{-3}$, consistent with Feingold et al. (1999). While this PE concentration falls within the range observed for giant sea-salt particles over the ocean (Jung et al., 2015), for clouds in a pristine environment, substantially higher numbers and sizes of PE are required to achieve effective precipitation acceleration. These observations are based on measurements of sea-salt aerosols (2–20 μm), which have large solution masses and corresponding large equilibrium sizes. However, under atmospheric conditions, these particles might not have sufficient time to grow to their equilibrium size (Ivanova et al., 1977), potentially resulting in lower PE concentrations. On the other hand, it is possible that PE concentrations can increase through stochastic collisions (Kostinski and Shaw, 2005; Dziekan and Pawlowska, 2017) as the cloud evolves. Furthermore, because the critical threshold decreases in DSDs with higher *collisional stability*, the effect of PEs is expected to be especially strong in non-precipitating, or polluted clouds as suggested in previous studies (e.g., Johnson, 1993; Dziekan et al., 2021).

While PEs can accelerate the rain initiation by collecting other droplets, they may reduce the number of raindrops by suppressing non-PE droplets to grow as raindrops. As a result, clouds without PEs may have more and larger raindrops, as PEs do not collect those before reaching the cloud top. This might lead to longer-lasting clouds and affect the precipitation differently. Thus, confirming this study's findings in more complex scenarios is necessary. Modeling efforts should incorporate additional processes such as aerosol activation, condensation, and entrainment. Especially, collisional droplet breakup (Low and List, 1982), is expected to increase the small number of PEs, causing more PE accretion afterward, and droplet sedimentation is expected to decrease the effect of PEs by making large raindrops precipitate and prevent PEs from further collisions

In conclusion, we confirm that a DSD barely producing raindrops is more sensitive to PEs (e.g., Dziekan et al., 2021). This underscores the need for caution in climate-engineering approaches like marine cloud brightening (Latham et al., 2012), aiming to create highly reflective clouds by artificially adding aerosol particles, where the unintended initiation of rain by adding large particles could be counterproductive (Hoffmann and Feingold, 2021). Indeed, this study found that PEs surpassing a critical threshold can initiate rain, while numerous PEs with a sufficiently small size are harmless. In addition, approaches to enhance precipitation, such as cloud seeding (Bowen, 1952; Cotton, 1982), should prioritize identifying target clouds with high stability and minimal rain production to maximize efficiency.

*Code and data availability.* A Python version of the LCM code is available on the link (https://github.com/jslim93/PyLCM_edu). The simulations were conducted using the FORTRAN version of the code, which employs the same collision routine as the Python version but provides faster computation. Simulation data will be made available upon request to the authors.

**Appendix A: Ensemble Size Sensitivity of $t_{100}$ and $t_{10\%}$**

Figure A1 illustrates how the mean and relative standard deviation of $t_{100}$ ($\mu_{100}$ and $\sigma_{100}$ / $\mu_{100}$, respectively) and $t_{10\%}$ ($\mu_{10\%}$ and $\sigma_{10\%}$ / $\mu_{10\%}$, respectively) evolve as the ensemble size increases from 1 to 200. Different colors of the dots represent RM8 (black), RM10 (blue), RM12 (red), RM14 (yellow) without PEs ($n_{\mathrm{PE}} = 0$). The mean values and the relative standard deviations converge when the ensemble size exceeds 100. Therefore, we consider an ensemble size of 100 adequate for obtaining reliable results.

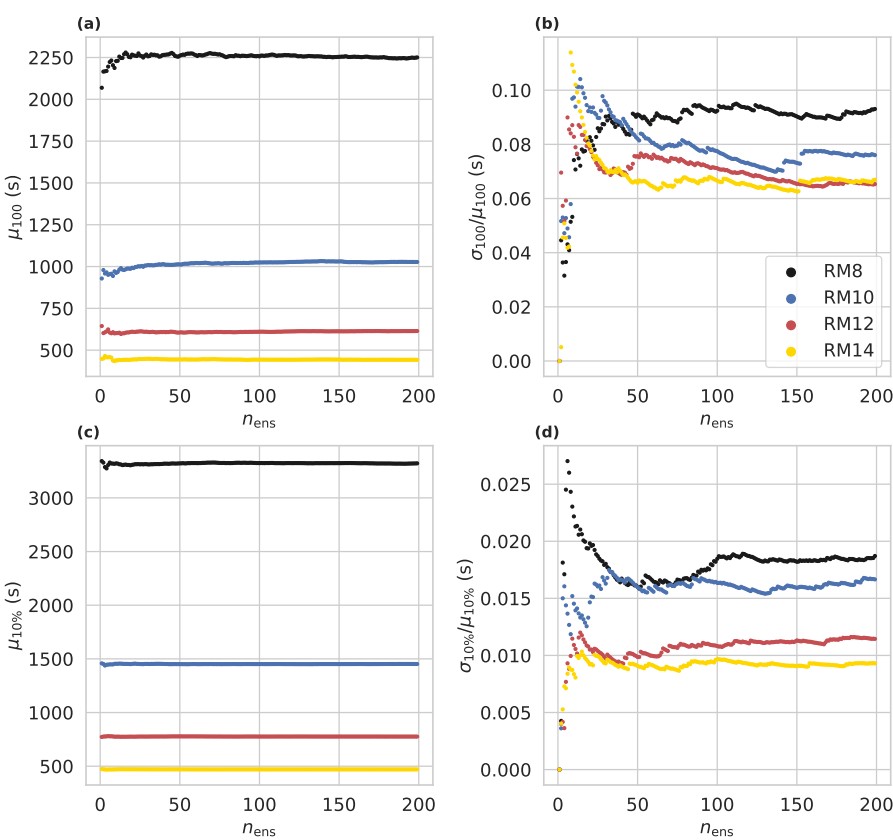

**Figure A1.** Variation of (a) $\mu_{100}$, (b) $\mu_{10\%}$, (c) $\sigma_{100}$ / $\mu_{100}$ and (d) $\sigma_{10\%}$ / $\mu_{10\%}$ with ensemble size ($n_{\mathrm{ens}}$) for RM8, RM10, RM12, and RM14 without PEs ($n_{\mathrm{PE}} = 0$).

## Appendix B: Parameters for the Fitting Function

Table B1 depicts the parameters and $r^2$ values derived from curve fitting Eq. (6) to $\mu_{100}$ for each result shown in Fig. 4a. Similarly, Table B2 shows the parameters and the $r^2$ values obtained from fitting Eq. (6) to $\mu_{10\%}$ for the results shown in Fig. 4b. The naming conventions for each case are as follows: Numbers following 'RM' denote $\bar{r}$ (e.g., 'RM8' corresponds to cases with $\bar{r} = 8$ μm). 'N' denotes cases with cut-off DSD. Numbers following 'T' indicate $\varepsilon$ (e.g., T50 corresponds to cases with $\varepsilon = 50$ cm$^2$ s$^{-3}$). The units of $\mu_{100,c}$ and $\mu_{10\%,c}$ are in s, $\Phi_{100}$ in μm$^{3.086}$ and $\Phi_{10\%}$ in μm$^{1.13}$. Units of $\Phi_{100}$ and $\Phi_{10\%}$ are determined by Eq. 5 with respective $a_\alpha$, $b_\alpha$ parameters, where the unit of $r_{PE}$ is μm and $n_{PE}$ is unit-less. The subscript $\alpha$ is 100 or 10% for $\mu_{100}$ and $\mu_{10\%}$, respectively. Thus, the units of $k_\alpha$ for $\mu_{100}$ and $\mu_{10\%}$ are μm$^{-3.086}$ s and μm$^{-1.13}$ s, respectively. In this study, these parameters are mainly used to compare how critical threshold varies in different cases than to obtain actual values.

**Table B1.** Parameters for fitting function of $\mu_{100}$

|  | RM8 | RM10 | RM12 | RM14 |
|---|---|---|---|---|
| $\Phi_{100,c}$ | $1.01 \times 10^4$ | $1.91 \times 10^4$ | $3.57 \times 10^4$ | $6.17 \times 10^4$ |
| $\mu_{100,c}$ | 2254.69 | 1025.24 | 609.73 | 435.49 |
| $k_{100}$ | 7.30 | 6.34 | 4.42 | 2.68 |
| $r^2$ | 0.99 | 0.99 | 0.98 | 0.95 |

|  | RM8N | RM10N | RM12N | RM14N |
|---|---|---|---|---|
| $\Phi_{100,c}$ | $1.00 \times 10^4$ | $1.57 \times 10^4$ | $1.93 \times 10^4$ | $2.04 \times 10^4$ |
| $\mu_{100,c}$ | 2275.34 | 1178.18 | 902.79 | 920.13 |
| $k_{100}$ | 7.30 | 6.13 | 5.81 | 5.70 |
| $r^2$ | 0.99 | 0.99 | 0.98 | 0.98 |

|  | RM10-T5 | RM10-T10 | RM10-T50 | RM10-T100 | RM10-T200 |
|---|---|---|---|---|---|
| $\Phi_{100,c}$ | $1.94 \times 10^4$ | $1.96 \times 10^4$ | $1.93 \times 10^4$ | $1.97 \times 10^4$ | $2.02 \times 10^4$ |
| $\mu_{100,c}$ | 998.16 | 984.67 | 888.95 | 798.79 | 687.37 |
| $k_{100}$ | 6.29 | 6.23 | 5.85 | 5.39 | 4.77 |
| $r^2$ | 0.99 | 0.99 | 0.98 | 0.96 | 0.94 |

**Table B2.** Parameters for fitting function of $\mu_{10\%}$

|  | RM8 | RM10 | RM12 | RM14 |
|---|---|---|---|---|
| $\Phi_{10\%,c}$ | $5.93 \times 10^1$ | $7.23 \times 10^1$ | $8.33 \times 10^1$ | $9.14 \times 10^1$ |
| $\mu_{10\%,c}$ | 3155.62 | 1405.09 | 763.82 | 466.49 |
| $k_{10\%}$ | 7.30 | 6.34 | 4.42 | 2.68 |
| $r^2$ | 0.75 | 0.83 | 0.88 | 0.88 |

|  | RM8N | RM10N | RM12N | RM14N |
|---|---|---|---|---|
| $\Phi_{10\%,c}$ | $5.93 \times 10^1$ | $6.86 \times 10^1$ | $7.27 \times 10^1$ | $7.30 \times 10^1$ |
| $\mu_{10\%,c}$ | 3160.14 | 1589.32 | 1186.52 | 1186.31 |
| $k_{10\%}$ | 7.30 | 6.13 | 5.81 | 5.70 |
| $r^2$ | 0.75 | 0.82 | 0.86 | 0.89 |

|  | RM10-T5 | RM10-T10 | RM10-T50 | RM10-T100 | RM10-T200 |
|---|---|---|---|---|---|
| $\Phi_{10\%,c}$ | $7.24 \times 10^1$ | $7.23 \times 10^1$ | $7.25 \times 10^1$ | $7.27 \times 10^1$ | $7.30 \times 10^1$ |
| $\mu_{10\%,c}$ | 1376.65 | 1352.81 | 1205.85 | 1067.95 | 889.78 |
| $k_{10\%}$ | 6.29 | 6.23 | 5.85 | 5.39 | 4.77 |
| $r^2$ | 0.82 | 0.82 | 0.81 | 0.80 | 0.77 |

*Author contributions.* JSL, YN, HL, and FH conceived the original conceptualization and interpretation of results and contributed to discussions. FH provided the base model and JSL modified the base model and formally analyzed the results. JSL wrote the original draft and JSL, YN, FH, and HL contributed to the review and editing. YN and FH provided the funding acquisition for the study and project administration.

*Competing interests.* There are no competing interests.

*Acknowledgements.* This work was supported by the Korea Meteorological Administration Research and Development Program under grant KMI2021-01512, the National Research Foundation of Korea (NRF) grant funded by the Korean government (MSIT) (2021R1F1A1051121), and the Emmy-Noether Program of the German Research Foundation (DFG) under grant HO 6588/1-1. The authors gratefully acknowledge the Gauss Centre for Supercomputing e.V. (www.gauss-centre.eu) for funding this project by providing computing time on the GCS Supercomputer SuperMUC-NG at Leibniz Supercomputing Centre (www.lrz.de). The authors greatly appreciate the comments of two anonymous reviewers that greatly improved the quality of the paper.

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
