# Peer review of "The Critical Number and Size of Precipitation Embryos to Accelerate Warm Rain Initiation"

_EGUsphere, 2024_

## Referee Comment (RC1)

**Referee's Comment on "The Critical Number and Size of Precipitation Embryos to Accelerate Warm Rain Initiation"**

Title: The Critical Number and Size of Precipitation Embryos to Accelerate Warm Rain Initiation
Author(s): Jung-Sub Lim et al.
MS No.: egusphere-2024-2636
MS type: Research article

**General Comments**

This paper offers a fresh perspective on the role that rare large droplets, here named "precipitation embryos (PEs)", can have on precipitation formation from a droplet distribution which would otherwise undergo slower collision-coalescence (akin to the "luck-drops" theory). The study is well-motivated and the use of the super-droplet model throughout is particularly appropriate. The conclusions drawn are well-grounded and are worthwhile for our understanding of the timing of rain formation. They certainly further our knowledge of how sufficiently numerous and large droplets can accelerate droplet collision-coalescence, and they also give particularly useful insight to how turbulence-induced collision enhancement (TICE) - another process known to accelerate rain formation - can either complement or inhibit the acceleration caused by large droplets. Indeed I believe the study to be within the scope of ACP and I recommend that the editor consider publishing this work. Nevertheless I have some specific comments I would like to see addressed first, particularly with regard to section 3 and the decision to truncate the droplet size distribution (DSD) for almost all the analysis.

**Specific Comments**

Points throughout the manuscript:

Most critically, I am unconvinced by the decision to truncate the DSD at $20\mu$m and I believe the analysis and conclusions of the paper would be clearer if they were presented without this cut-off. The truncation is justified as creating a "stable initial DSD in which collisions are negligible" however, as for example the black lines in figure 6 show, the truncated DSDs do undergo frequent collisions and so the truncation appears unjustified. Contrary to the Author's introduction, I believe such examples also mean the role of "precipitation embryos" (PEs) in inducing collisions in an otherwise stable distribution is not addressed in this paper. The paper is better introduced as showing how PEs can accelerate the timing of rain formation for a DSD in which droplets *already can* collide. I am of the opinion that the analysis would be clearer, and the conclusions more convincing, if the cut-off was removed. For example the red and green lines in figures 4(a) and 4(b) (i.e. the RM12 and RM14 setups) show large changes between figures 4(a) and 4(b), and 4(c) and 4(d), ie. when the un-truncated distribution is used instead. Likewise figure 5 shows large differences in the results without truncation. This means the behaviour of the critical threshold is heavily influenced by the cut-off at $20\mu$m and the analysis of the role of PEs, particularly in section 3, would be more convincingly shown if that influence wasn't present.

*Points regarding certain lines within the manuscript:*

- L3 (and repeated): please reconsider the use of the term "colloidally stable". The terminology is misleading because it is conventionally used to discuss the condition of a colloid substance - i.e. the mixture of soluble and insoluble substances in a solute.

- L22 ("so-called precipitation embryos (PEs)"): is the term "precipitation embryo" coined in this paper? If not, please provide a citation here. If so, please consider if there is any existing terminology you can use instead. Why for example is the term "lucky droplet" not appropriate?

- L81 ("$N_0 = 238$ cm$^{-3}$, 456 cm$^{-3}$ and 523 cm$^{-3}$"): the values for $N_0$ given here do not match those in the legend of figure 1 so please correct them. Also it would be clearer if their order was consistent with the ordering of $\bar{r}$ in the sentence before (i.e. if $\bar{r}$ is ascending, $N_0$ is descending).

- Figure 1: These DSDs do not show very clearly the alteration you make by truncating these distributions, nor do they show the impact of adding PEs on them. It would be helpful to the reader to include a neighbouring figure (or to replace this one) with a plot that shows the truncated DSDs and some DSDs which example the distribution with $n_{\mathrm{PE}}, r_{\mathrm{PE}}$ values.

- L96 ("To explore the impact of PEs, we investigate 49 ensemble simulations…"): it's quite important that this is justified with a comparison to the real world. How do these PE radii and number concentrations compare with observations? Especially relevant would be to put them in the context of shallow convective clouds, since that's the range of TICE you consider later on, or cloud seeding. Context is particularly necessary because figure 2 shows both $\mu_{100}$ and $\mu_{10\%}$ have almost no dependency on a large portion of the $n_{\mathrm{PE}}, r_{\mathrm{PE}}$ space spanned by the ensemble.

- L102 ("larger PEs can substantially increase the initial $q_c$"): please state how much your simulations are perturbed by the presence of PEs. For example stating what the largest $q_c$ of your ensemble is (when $(n_{\mathrm{PE}}, r_{\mathrm{PE}}) = (1000, 18)$).

- L213 ("although $t_{100}$ can be shorter than in the cases without PEs."): please provide a suggested explanation or citation for this statement.

- Figure 6: With increasing PE concentration, the accretion and auto-conversion rates due to *non-PE* collisions maximise earlier. A acknowledgement and a possible reason for this behaviour would be beneficial because it is not self-evident why it should occur.

**Technical Corrections**

*Points regarding certain lines within the manuscript:*

- L11 ("A key question in warm rain initiation is to explain..."): There is no question posed here, consider rephrasing the second half of the sentence into a question, or rewording "a key question".

- L36 ("In particular, if we consider ... Thus, it is also important to account for stochastic fluctuations in the collision process."): please rephrase this sentence because this logic is hard to follow. I was left asking myself: why is a 12.5$\mu$m droplet relevant to your study, and how does this citation justify the need for stochastic fluctuations? I assume you mean than by including stochasticity you don't need as large lucky droplets to initiate collisions, but please make this more explicit if so.

- L56 ("frequently called superdroplets by introducing a weighting factor"): please add missing comma after "superdroplets".

- L110 ("The timescale t_10% represents the time when 10 % of the initial cloud droplet mass converts to rain…"): is rain here defined as above 40 or 100 microns?

- L145 (equation 6): Missing subscript $\alpha$ on $\Phi$ on the left-hand side of the equation.

- Figure 4 and figure 5: Please reconsider the colours you use to plot these figures because they are unkind to colour-blindness (especially figure 4). Also in figure 4, the re-use of the same colours for subplots (e) and (f), although they are no longer denoting different $\bar{r}$, is undesirable.

---

## Referee Comment (RC2)

**Review of "The Critical Number and Size of Precipitation Embryos to Accelerate Warm Rain Initiation" by Lim et al.**

This paper presents a numerical investigation into the role of low-concentration, large-sized droplets, referred to as precipitation embryos (PEs), in the formation of rain in warm clouds. The study employs box simulations focused exclusively on the process of collision-coalescence. The authors utilize a "one-to-one" Lagrangian cloud model (LCM), which is an appropriate choice for capturing the detailed mechanisms of collision-coalescence. By modeling small cloud volumes (less than one cubic meter) and excluding other in-cloud processes, the study provides qualitative insights into the role of PEs in rain initiation. However, this narrow focus allows the authors to explore a broad range of droplet sizes and concentrations, as well as the effects of turbulence-induced collision enhancement (TICE).

The results of the study corroborate previously observed phenomena, such as the competition between PEs and TICE, and the particularly significant role of PEs in clouds that otherwise would not precipitate. While the findings themselves may not be particularly groundbreaking, they contribute to a deeper understanding of warm-rain formation. Notably, the paper introduces a useful formalism for determining the minimum number and size of PEs required to influence rain formation. I believe this study is a valuable contribution to the field and would recommend its publication, provided the authors address the following points:

1. **Clarification of Non-PE Concentrations**: It is unclear what non-PE concentrations were used. Based on line 81, these appear to be 238, 456, and 523 droplets per cubic centimeter, but the caption of Figure 1 provides conflicting information. It would also be beneficial to ensure that there is a case representing a clear marine concentration of approximately 100 droplets per cubic centimeter, given that sea-spray aerosols are a known source of PEs.

2. **Range of Dissipation Rates for TICE**: The paper tests TICE for dissipation rates of 16, 80, and 100 cm²/s³. However, the values of 80 and 100 cm²/s³ are quite similar, and all the tested rates fall within the typical range for cumulus clouds (10–100 cm²/s³). It would be more informative to include dissipation rates representative of other cloud types, such as stratocumulus, cumulus, and deep convection, to broaden the study's applicability.

3. **Discussion of PE Characteristics in Real Clouds**: The paper would benefit from a discussion of the expected number and size of different types of PEs in real clouds, along with an assessment of their likely significance in natural cloud systems.

4. **Interpretation of Function ϕ in Equation (5)**: The function ϕ, defined in Equation (5), describes the impact of PEs by combining their number and size. However, it is a decreasing function of PE size and concentration, which makes the plots harder to interpret. It might be more intuitive if ϕ were an increasing function, and if ϕ = 0 corresponded to the absence of PEs.

5. **Definition of Rain in the 10% Mass Conversion**: It is unclear how rain is defined when calculating the time required to convert 10% of the total cloud mass to rain. Is the initial presence of rainwater (e.g., from PEs already exceeding the rain threshold at t = 0 s) included in the 10% mass calculation?

6. **Interpretation of Results Regarding the 'Lucky Droplet' Theory**: The statement on line 190 —"This suggests that while PEs can accelerate the formation of the largest raindrops, these droplets may not substantially impact overall rain initiation after the initial period when they are few"—is described as contradicting the 'lucky droplet' theory. However, I disagree with this interpretation. The results show that when there are too few lucky droplets (PEs), they do not significantly impact the time to convert 10% of the cloud mass into rain. Nonetheless, when more lucky droplets are present, they do affect this time ($t_{10\%}$). Furthermore, even if $t_{10\%}$ averaged over small cloud volumes is not affected, the presence of lucky droplets could still play an important role in rain formation in real clouds.

**Technical Comments:**

- Line 135: There is an unclosed parenthesis.

---

## Author Comment (AC1)

**Authors' Response to Reviews of**

**The Critical Number and Size of Precipitation Embryos to Accelerate Warm Rain Initiation**

Jung-Sub Lim, Yign Noh, Hyunho Lee, Fabian Hoffmann
*Atmospheric Chemistry and Physics,*
* * *
RC: *Reviewers' Comment*,    AR: Authors' Response,    ☐ Manuscript Text

*All line numbers in this response letter refer to the revised manuscript unless stated.*

RC: *This paper offers a fresh perspective on the role that rare large droplets, here named "precipitation embryos (PEs)", can have on precipitation formation from a droplet distribution which would otherwise undergo slower collision-coalescence (akin to the "luck-drops" theory). The study is well-motivated and the use of the superdroplet model throughout is particularly appropriate. The conclusions drawn are well-grounded and are worthwhile for our understanding of the timing of rain formation. They certainly further our knowledge of how sufficiently numerous and large droplets can accelerate droplet collision-coalescence, and they also give particularly useful insight to how turbulence-induced collision enhancement (TICE) - another process known to accelerate rain formation - can either complement or inhibit the acceleration caused by large droplets. Indeed I believe the study to be within the scope of ACP and I recommend that the editor consider publishing this work. Nevertheless I have some specific comments I would like to see addressed first, particularly with regard to section 3 and the decision to truncate the droplet size distribution (DSD) for almost all the analysis.*

AR: Dear reviewer, we highly appreciate your effort and time to evaluate this manuscript and provide fruitful comments that greatly improved the quality of the paper. We have revised our manuscript accordingly. Below you will find a point-by-point response from the authors.

**1.  Comment 1**

RC: *Most critically, I am unconvinced by the decision to truncate the DSD at 20 m and I believe the analysis and conclusions of the paper would be clearer if they were presented without this cut-off. The truncation is justified as creating a "stable initial DSD in which collisions are negligible" however, as for example the black lines in figure 6 show, the truncated DSDs do undergo frequent collisions and so the truncation appears unjustified. Contrary to the Author's introduction, I believe such examples also mean the role of "precipitation embryos" (PEs) in inducing collisions in an otherwise stable distribution is not addressed in this paper. The paper is better introduced as showing how PEs can accelerate the timing of rain formation for a DSD in which droplets already can collide. I am of the opinion that the analysis would be clearer, and the conclusions more convincing, if the cut-off was removed. For example the red and green lines in figures 4(a) and 4(b) (i.e. the RM12 and RM14 setups) show large changes between figures 4(a) and 4(b), and 4(c) and 4(d), ie. when the un-truncated distribution is used instead. Likewise figure 5 shows large differences in the results without truncation. This means the behaviour of the critical threshold is heavily influenced by the cutoff at 20 m and the analysis of the role of PEs, particularly in section 3, would be more convincingly shown if that influence wasn't present.*

AR: We also agree with the reviewer that DSD without cut-off represents a more natural DSD. The original

motivation for the cut-off was to isolate the effects of PEs by starting from an initially *stable* DSD, in which droplets are not large enough to initiate frequent collisions. However, as the reviewer correctly pointed out, collisions can still occur in the truncated DSDs, even without PEs. To address this, we have revised the manuscript by making the case without the DSD cut-off (*broad cases*) as the base case and renamed the cases with DSD cut-off as *narrow* cases.

AR:   In particular, Sec. 3 now shows the results without the DSD cut-off. The *narrow* cases are only used to discuss how the PE critical value changes with DSD shape, particularly with respect to the largest initial droplet size. We note that in cases where $\bar{r} = 10~\mu m$ (light blue lines in Fig.1), there are very few cloud droplets larger than $20 \mu m$ even without the DSD cut-off. Thus, the conclusions in Sec. 3 are not substantially different from the original manuscript.

**2. Comment 2**

RC:   *L3 (and repeated): please reconsider the use of the term "colloidally stable". The terminology is misleading because it is conventionally used to discuss the condition of a colloid substance - i.e. the mixture of soluble and insoluble substances in a solute.*

AR:   Although Squires (1958) originally used the term "colloidal stability" to describe the ability of a DSD to initiate precipitation, we agree with the reviewer that this term has a different conventional meaning in other areas of the scientific community. Therefore, we decided to use the term *collisional stability*, and defined it in the introduction.

> **L16:** These collision-limited DSDs can be regarded as being in a *collisionally stable* state (Squires, 1958), where mechanisms that accelerate the collision-coalescence process to form raindrops and initiate precipitation are crucial for breaking this stability.

**3. Comment 3**

RC:   *L22 ("so-called precipitation embryos (PEs)"): is the term "precipitation embryo" coined in this paper? If not, please provide a citation here. If so, please consider if there is any existing terminology you can use instead. Why for example is the term "lucky droplet" not appropriate?*

AR:   The term precipitation embryos was originally used to describe large droplets that initiate warm rain in earlier studies (e.g., Pontikis et al., 1987; Johnson, 1993). In this study, we focus on the effects of large droplets that initiate warm rain, regardless of their origin—whether from giant aerosol particles, "lucky droplets" arising from stochastic collisions, or seeded particles. While the term "lucky droplet" implies a droplet form from a series of unlike stochastic collisions, and "giant aerosol" refers to initially large droplets, we find that the broader concept of precipitation embryos which encompasses large droplets formed through multiple potential mechanisms, most suitable for this study. We have cited Johnson (1993), as it provides a definition most closely aligned with the concept used in our research.

> **L24:** ... and (iii) the role of so-called precipitation embryos (PEs; e.g., Johnson, 1993), the primary focus of this study.

**4. Comment 4**

**RC:** *L81 ("$N_0$ = 238 cm-3, 456 cm-3 and 523 cm-3"): the values for $N_0$ given here do not match those in the legend of figure 1 so please correct them. Also it would be clearer if their order was consistent with the ordering of in the sentence before (i.e. if $\bar{r}$ is ascending, $N_0$ is descending).*

**AR:** Thank you for pointing this out. We have corrected the values in the text to match those in the legend of Figure 1, which were correct. We also reordered the values to ensure that when $\bar{r}$ is ascending, $N_0$ is descending, as suggested.

> **L92:** In these cases, $N_0 = 466,\ 138,\ \text{and}\ 87\ \text{cm}^{-3}$ to achieve the same $q_c = 1.0\ \text{g kg}^{-1}$ (Fig. 1).

**5. Comment 5**

**RC:** *Figure 1: These DSDs do not show very clearly the alteration you make by truncating these distributions, nor do they show the impact of adding PEs on them. It would be helpful to the reader to include a neighbouring figure (or to replace this one) with a plot that shows the truncated DSDs and some DSDs which example the distribution with $n_{\mathrm{PE}}, r_{\mathrm{PE}}$ values.*

**AR:** Thank you for the recommendation. We have incorporated a second panel in Figure 1 to address your suggestion. This panel now illustrates the truncated DSDs and includes an example PE distribution with $r_{\mathrm{PE}} = 27\ \mu\mathrm{m}$ and different $N_{\mathrm{PE}}$ values to provide a comprehensive representation of the DSD cut-off and PE addition.

[Figure]

Figure 1: (a) Initial DSDs for various $\bar{r}$ and their corresponding $N_0$ values. The dotted line indicates the cut-off radius of $20\ \mu\mathrm{m}$, above which droplets are removed in cases with a DSD cut-off. (b) Initial DSDs with a DSD cut-off for various $\bar{r}$ and $N_0$ values, along with a vertical bar plot showing various PE distributions for $r_{\mathrm{PE}} = 27\ \mu\mathrm{m}$ and various $n_{\mathrm{PE}}$ values. **Excerpted from Fig. 1 of the manuscript.**

**6. Comment 6**

**RC:** *L96 ("To explore the impact of PEs, we investigate 49 ensemble simulations...."): it's quite important that this is justified with a comparison to the real world. How do these PE radii and number concentrations compare with observations? Especially relevant would be to put them in the context of shallow convective clouds, since that's the range of TICE you consider later on, or cloud seeding. Context is particularly necessary because figure 2 shows both $\mu_{100}$ and $\mu_{10\%}$ have almost no dependency on a large portion of the $n_{\mathrm{PE}}, r_{\mathrm{PE}}$ space spanned by the ensemble.*

**AR:** We agree with the reviewer that providing the observed range of $n_{\mathrm{PE}}$ and $r_{\mathrm{PE}}$ values is essential for context. In the revised manuscript, we have included references to observed PE number concentrations across different cloud types. Notably, the smallest $n_{\mathrm{PE}} = 1$ is lower than typically observed PE concentrations. However, this value was included to explore the "lucky droplet" effect, where a single "one-in-a-million" droplet is expected to accelerate the collision process. Now we relate the PE concentrations considered in this study to the observed values over the ocean to incorporate the reviewer's comment.

**AR:** In addition, we have excluded cases with $r_{\mathrm{PE}} = 15\ \mu$m in the revised manuscript to improve clarity. While these cases were initially included to demonstrate the limited effect of small PEs ($r_{\mathrm{PE}} < 20\ \mu$m), the effect is sufficiently represented by the cases with $r_{\mathrm{PE}} = 18\ \mu$m making the discussion clearer.
* * *
**L45:** Lastly, there is a large uncertainty in the number concentration of PEs in clouds (Khain, 2009). For instance, PEs originating from $1$–$20\ \mu$m sea salt aerosols exhibit a wide range of concentrations from $10^{-4}$ to $10^{-2}\ \mathrm{cm}^{-3}$ (Jung et al., 2015; Jensen & Nugent, 2017), with a strong environmental and spatial dependency (Woodcock, 1953; Jung et al., 2015). Based on the "one in a million" definition of "lucky droplet" acting as PEs (e.g., Kostinski & Shaw, 2005), typical cloud droplet concentrations over the ocean and continents ($10^1$ to $10^3\ \mathrm{cm}^{-3}$) imply PE concentrations of $10^{-5}$ to $10^{-3}\ \mathrm{cm}^{-3}$. On the other hand, for climate-engineering practices such as cloud seeding, the concentration of seeded particles can exceed natural values, ranging from $10^{-1}$ to $10^1\ \mathrm{cm}^{-3}$ (Kuba & Murakami, 2010). Due to this large variability, assessing the PE effect for a broad range of PE concentrations is important.

**L110:** We choose a minimum $n_{\mathrm{PE}} = 1$ to investigate whether a 'one in a million' PE can accelerate droplet collision, as highlighted in previous studies on lucky droplets (Kostinski & Shaw, 2005; Dziekan & Pawlowska, 2017). Within a given reference volume, the minimum and maximum $n_{\mathrm{PE}}$ values of 1 and 1000 correspond to concentrations of approximately $2.97 \times 10^{-4}\ \mathrm{cm}^{-3}$ and $2.97 \times 10^{-1}\ \mathrm{cm}^{-3}$, respectively, reflecting the wide range of PE concentrations observed in nature (Khain, 2009; Jung et al., 2015).

**L146:** For RM10, when $n_{\mathrm{PE}} = 3$, the PE number concentration is approximately $10^{-3}\ \mathrm{cm}^{-3}$. In this case, even PEs larger than $40\ \mu$m are not effective in accelerating $t_{10\%}$ (Fig. 2b). However, when the PE concentration increases to a relatively high value ($n_{\mathrm{PE}} = 30$), which corresponds to the maximum value rarely observed over the ocean (Jung et al., 2015), PEs larger than $22\ \mu$m can substantially accelerate $t_{10\%}$ (Fig. 2b). In contrast, for RM14, which represents typical maritime clouds in a pristine environment with $N_0 = 87\ \mathrm{cm}^{-3}$, the effect of PEs becomes substantially less effective. PEs smaller than $33\ \mu$m are unable to accelerate $\mu_{100}$ regardless of $n_{\mathrm{PE}}$ (Fig. 2c). Moreover, only cases with both very large $n_{\mathrm{PE}}$ and $r_{\mathrm{PE}}$ can accelerate $t_{10\%}$ (Fig. 2d). This implies that the PE effect also depends on the initial DSD shape, making it necessary to have a stable, non-precipitating cloud for the PE effect to be substantial. However, such extreme conditions are unlikely to occur in typical maritime environments. This highlights that the PE effect strongly depends on the initial DSD shape and requires

stable, non-precipitating cloud conditions to be substantial.

**L308:** In this study, PEs larger than 22 $\mu$m are found to effectively accelerate the precipitation ($t_{10\%}$) for clouds in relatively polluted environments, when their concentration exceeds $10^{-3}$ cm$^{-3}$, consistent with Feingold et al. (1999). While this PE concentration falls within the range observed for giant sea-salt particles over the ocean (Jung et al., 2015), for clouds in a pristine environment, substantially higher numbers and sizes of PE are required to achieve effective precipitation acceleration. These observations are based on measurements of sea-salt aerosols (2–20 $\mu$m), which have large solution masses and corresponding large equilibrium sizes. However, under atmospheric conditions, these particles might not have sufficient time to grow to their equilibrium size (Ivanova et al., 1977), potentially resulting in lower PE concentrations. On the other hand, PE concentrations can also increase through stochastic collisions (Kostinski & Shaw, 2005; Dziekan & Pawlowska, 2017) suggesting their concentration can continue to grow as the cloud evolves. Furthermore, because the critical threshold decreases in DSDs with higher colloidal stability, the effect of PEs is expected to be especially strong in non-precipitating, or polluted clouds as suggested in previous studies (e.g., Johnson, 1993; Dziekan et al., 2021).

**7. Comment 7**

**RC:** *L102 ("larger PEs can substantially increase the initial $q_c$"): please state how much your simulations are perturbed by the presence of PEs. For example stating what the largest $q_c$ of your ensemble is (when $n_{\mathrm{PE}}, r_{\mathrm{PE}}$ = 1000,18 )*

**AR:** Thank you for this comment. In response, we have quantified the perturbations caused by the addition of PEs. The maximum increases in $q_c + q_r$ are 12.5% and 6.4%, respectively, which were excluded from the original analysis. To address this, we have now introduced a criterion whereby only cases are included in the analysis in which $q_c + q_r$ increase less than 2% due to the addition of PEs (see the values in Fig. 2 of this letter). For most cases, the increase in $q_c + q_r$ caused by PEs is below 1%.

**L130:** Adding PEs increases the initial $q_c$, or the rainwater mixing ratio $q_r$ when $r_{\mathrm{PE}} > 40$ $\mu$m and $n_{PE} > 0$, potentially limiting the comparability of simulated cases. To address this, we restricted the analysis of $t_{10\%}$ and further conversion rates such as autoconversion rate (i.e., raindrop formation by collisions between cloud droplets), and accretion rate (i.e., raindrop growth by raindrops collecting cloud droplets) to cases where the increase in the initial $q_c + q_r$ due to the addition of PEs is below 2 %. In most cases, the increase in $q_c$ and $q_r$ is below 1 %. However, two exceptions, $n_{\mathrm{PE}} = 300$, with $r_{\mathrm{PE}} = 40$ $\mu$m and $n_{\mathrm{PE}} = 1000$, with $r_{\mathrm{PE}} = 27$ $\mu$m, show an increase of 1.9 %.

**8. Comment 8**

**RC:** *L213 ("although $t_{100}$ can be shorter than in the cases without PEs."): please provide a suggested explanation or citation for this statement.*

**AR:** This sentence refers that $t_{100}$ is shorter in the cases with small $n_{\mathrm{PE}}$ values compared to the $t_{100}$ value in the case without PEs. We acknowledge the reviewer's concern about the ambiguity and changed the sentence accordingly.

[Figure]

Figure 2: Heatmap of change in initial $q_c + q_r$ by PEs for different $r_{\mathrm{PE}}$ and $n_{\mathrm{PE}}$ values.

> **L246:** Thus, although larger PEs are more likely to collide, the overall collision frequency remains low when $n_{\mathrm{PE}}$ is small, resulting in slower PE autoconversion compared to non-PE autoconversion. While non-PE-autoconversion always decreases with increasing $n_{\mathrm{PE}}$, PE-autoconversion increases substantially only when $n_{\mathrm{PE}} \geq 100$. Therefore, before exceeding the critical threshold, PEs suppress non-PE autoconversion more than they enhance autoconversion which can even lead to a decrease in the total (PE and non-PE) autoconversion. Hence, shorter $t_{100}$ does not necessarily lead to a shorther $t_{10\%}$ when $n_{\mathrm{PE}}$ is small (Fig. 2).

**9. Comment 9**

**RC:** *Figure 6: With increasing PE concentration, the accretion and auto-conversion rates due to non-PE collisions maximise earlier. A acknowledgement and a possible reason for this behaviour would be beneficial because it is not self-evident why it should occur.*

**AR:** This behavior occurs because PE particles suppress non-PE autoconversion, as discussed in this paper. As shown in Fig. 6c, the non-PE autoconversion rate is nearly identical across all cases during the initial 1000–1200 s, regardless of PE concentration. However, when the PE effect is stronger, more non-PE droplets are collected by PEs, reducing the number of droplets available for autoconversion. As a result, in cases with higher PE concentrations, the autoconversion rate peaks earlier and then begins to decline. This suppression of non-PE autoconversion also decreases the number of non-PE raindrops, which reduces the non-PE accretion rate and causes it to peak earlier in these cases. These findings suggest that the primary role of PEs is to collect non-PE droplets, while indirectly suppressing both non-PE autoconversion and accretion. We have included this explanation in the revised manuscript.

> **L265:** Interestingly, at high $n_{\mathrm{PE}}$, the non-PE autoconversion and accretion rates reach their peak values earlier than in cases without PEs or with low $n_{\mathrm{PE}}$ (Fig. 6c, d, e, and f). During the initial 1000 s, the non-PE autoconversion rate is nearly identical across all cases, regardless of $n_{\mathrm{PE}}$. However, when $n_{\mathrm{PE}}$ is high, more non-PE droplets are collected by PEs, reducing the number of droplets available for autoconversion. As a result, the non-PE autoconversion rate peaks and declines earlier in cases with higher PE concentrations. This suppression of non-PE autoconversion decreases the number of non-PE raindrops and the non-PE accretion rate. These findings highlight that the primary role of PEs is to collect non-PE droplets, hence suppressing non-PE autoconversion and accretion.

**10. Technical Comments**

**RC:** *L11 ("A key question in warm rain initiation is to explain..."): There is no question posed here, consider rephrasing the second half of the sentence into a question, or rewording "a key question".*

AR: Done. Now, we address this as a key challenge.

**RC:** *L36 ("In particular, if we consider ... Thus, it is also important to account for stochastic fluctuations in the collision process."): please rephrase this sentence because this logic is hard to follow. I was left asking myself: why is a 12.5 $\mu m$ droplet relevant to your study, and how does this citation justify the need for stochastic fluctuations? I assume you mean than by including stochasticity you don't need as large lucky droplets to initiate collisions, but please make this more explicit if so.*

AR: We agree with the reviewer that the reference to a 12.5 $\mu m$ droplet is not directly relevant to our study. The original intention was to highlight that including stochasticity reduces the need for larger or more PEs to initiate rain. To address the reviewer's concerns, we have rephrased the sentence to clarify this point and provide a better context for the importance of collision stochasticity.

> **L39:** In the absence of PEs, DSDs with small-sized droplets barely initiate precipitation unless stochastic fluctuations in the collision process are considered. This phenomenon is known as the "lucky droplet" effect, which may produce PEs on its own (Telford, 1955; Kostinski & Shaw, 2005; Dziekan & Pawlowska, 2017). When this effect dominates, adding only a few PEs may not substantially accelerate rain initiation.

**RC:** *L56 ("frequently called superdroplets by introducing a weighting factor"): please add missing comma after "superdroplets".*

AR: Done

**RC:** *L110 ("The timescale $t_{10\%}$ represents the time when $10\%$ of the initial cloud droplet mass converts to rain..."): is rain here defined as above 40 or 100 microns?*

AR: A raindrop is defined as a droplet larger than 40 $\mu$m, in radius in our study. We clarified this in the revised manuscript.

**RC:** *L145 (equation 6): Missing subscript on $\alpha$ on $\Phi$ the left-hand side of the equation.*

AR: Done.

**RC:** *Figure 4 and figure 5: Please reconsider the colours you use to plot these figures because they are unkind*

*to colour-blindness (especially figure 4). Also in figure 4, the re-use of the same colours for subplots (e) and (f), although they are no longer denoting different $\bar{r}$, is undesirable.*

AR: Thank you for your comment. We have now revised all figures to be accessible for color vision deficiency. Also, we assigned new colors to cases with the same $\bar{r}$ but different $\varepsilon$ in panels e and f of Figure 4, ensuring that these colors do not overlap with those used in panels a–d.

**References**

Dziekan, P., Jensen, J. B., Grabowski, W. W., & Pawlowska, H. (2021). Impact of giant sea salt aerosol particles on precipitation in marine cumuli and stratocumuli: Lagrangian cloud model simulations. *Journal of the Atmospheric Sciences*, *78*(12), 4127–4142. doi:

Dziekan, P., & Pawlowska, H. (2017). Stochastic coalescence in lagrangian cloud microphysics. *Atmospheric Chemistry and Physics*, *17*(22), 13509–13520. doi:

Feingold, G., Cotton, W. R., Kreidenweis, S. M., & Davis, J. T. (1999). The impact of giant cloud condensation nuclei on drizzle formation in stratocumulus: Implications for cloud radiative properties. *Journal of the atmospheric sciences*, *56*(24), 4100–4117. doi:

Ivanova, E., Kogan, Y., Mazin, I., & Permyakov, M. (1977). The ways of parameterization of condensation drop growth in numerical models. *Izv. Atmos. Oceanic Phys*, *13*, 1193–1201.

Jensen, J. B., & Nugent, A. D. (2017). Condensational growth of drops formed on giant sea-salt aerosol particles. *Journal of Atmospheric Sciences*, *74*(3), 679–697. doi:

Johnson, D. B. (1993). The onset of effective coalescence growth in convective clouds. *Quarterly Journal of the Royal Meteorological Society*, *119*(513), 925–933.

Jung, E., Albrecht, B. A., Jonsson, H. H., Chen, Y.-C., Seinfeld, J. H., Sorooshian, A., . . . Russell, L. M. (2015). Precipitation effects of giant cloud condensation nuclei artificially introduced into stratocumulus clouds. *Atmospheric Chemistry and Physics*, 5645–5658.

Khain, A. (2009). Notes on state-of-the-art investigations of aerosol effects on precipitation: a critical review. *Environmental Research Letters*, *4*(1), 015004.

Kostinski, A. B., & Shaw, R. A. (2005). Fluctuations and luck in droplet growth by coalescence. *Bulletin of the American Meteorological Society*, *86*(2), 235-244. doi:

Kuba, N., & Murakami, M. (2010). Effect of hygroscopic seeding on warm rain clouds–numerical study using a hybrid cloud microphysical model [Journal Article]. *Atmospheric Chemistry and Physics*, *10*(7), 3335-3351.

Pontikis, C., Isaka, H., Jochum, A., Jonas, P., & Schaller, E. (1987). Workshop on warm convective clouds, 9–10 february 1987, paris, france. *Bulletin of the American Meteorological Society*, *68*(10), 1254–1256.

Squires, P. (1958). The microstructure and colloidal stability of warm clouds: Part i—the relation between structure and stability. *Tellus*, *10*(2), 256–261.

Telford, J. (1955). A new aspect of coalescence theory. *Journal of Meteorology*, *12*(5), 436-444. doi:

Woodcock, A. H. (1953). Salt nuclei in marine air as a function of altitude and wind force. *Journal of Atmospheric Sciences*, *10*(5), 362–371. doi:

---

## Author Response (AR1)

**Authors' Response to Reviews of**

**The Critical Number and Size of Precipitation Embryos to Accelerate Warm Rain Initiation**

Jung-Sub Lim, Yign Noh, Hyunho Lee, Fabian Hoffmann
*Atmospheric Chemistry and Physics,*
* * *
RC: *Reviewers' Comment*,     AR: Authors' Response,     ☐ Manuscript Text

*All line numbers in this response letter refer to the revised manuscript unless stated.*

RC: *This paper presents a numerical investigation into the role of low-concentration, large-sized droplets, referred to as precipitation embryos (PEs), in the formation of rain in warm clouds. The study employs box simulations focused exclusively on the process of collision-coalescence. The authors utilize a "one-to-one" Lagrangian cloud model (LCM), which is an appropriate choice for capturing the detailed mechanisms of collision-coalescence. By modeling small cloud volumes (less than one cubic meter) and excluding other in-cloud processes, the study provides qualitative insights into the role of PEs in rain initiation. However, this narrow focus allows the authors to explore a broad range of droplet sizes and concentrations, as well as the effects of turbulence-induced collision enhancement (TICE). The results of the study corroborate previously observed phenomena, such as the competition between PEs and TICE, and the particularly significant role of PEs in clouds that otherwise would not precipitate. While the findings themselves may not be particularly groundbreaking, they contribute to a deeper understanding of warm-rain formation. Notably, the paper introduces a useful formalism for determining the minimum number and size of PEs required to influence rain formation. I believe this study is a valuable contribution to the field and would recommend its publication, provided the authors address the following points:*

AR: Dear Reviewer, we sincerely appreciate your time and effort in reviewing our manuscript and providing valuable feedback, which has helped clarify the discussion. We have revised the manuscript accordingly. Below you will find a point-by-point response from the authors.

**1. Comment 1**

RC: *Clarification of Non-PE Concentrations: It is unclear what non-PE concentrations were used. Based on line 81, these appear to be 238, 456, and 523 droplets per cubic centimeter, but the caption of Figure 1 provides conflicting information. It would also be beneficial to ensure that there is a case representing a clear marine concentration of approximately 100 droplets per cubic centimeter, given that sea-spray aerosols are a known source of PEs.*

AR: The text in line 81 contained a misprint and the number concentrations provided in the label of Figure 1 are correct. We changed the text accordingly. Note that the, two cases with $N_0 = 138$ and $87 \text{ cm}^{-3}$ and $\bar{r} = 12, 14 \ \mu\text{m}$, represent typical clean maritime conditions.

> **L92:** In these cases, $N_0 = 466, \ 138, \text{ and } 87 \text{ cm}^{-3}$ to achieve the same $q_\text{c} = 1.0 \text{ g kg}^{-1}$ (Fig. 1).

**2. Comment 2**

**RC:** *Range of Dissipation Rates for TICE: The paper tests TICE for dissipation rates of 16, 80, and 100* $\mathrm{cm^2\,s^{-3}}$*. However, the values of 80 and 100* $\mathrm{cm^2\,s^{-3}}$ *are quite similar, and all the tested rates fall within the typical range for cumulus clouds (10–100* $\mathrm{cm^2\,s^{-3}}$*). It would be more informative to include dissipation rates representative of other cloud types, such as stratocumulus, cumulus, and deep convection, to broaden the study's applicability.*

**AR:** Now we change the considered $\varepsilon$ values to 5, 10, 50, 100, 200 $\mathrm{cm^2\,s^{-3}}$ which accounts for typical stratocumulus, shallow cumulus, and deep convective clouds.

> **L118:** To investigate the effect of TICE, five different kinetic energy dissipation rates $\varepsilon = 5,\ 10,\ 50,\ 100,$ and $200\,\mathrm{cm^2\,s^{-3}}$ are considered for RM10.

**3. Comment 3**

**RC:** *Discussion of PE Characteristics in Real Clouds: The paper would benefit from a discussion of the expected number and size of different types of PEs in real clouds, along with an assessment of their likely significance in natural cloud systems.*

**AR:** We appreciate this comment. We now discuss PE concentrations compared to the atmospheric conditions:

> **L45:** Lastly, there is a large uncertainty in the number concentration of PEs in clouds (Khain, 2009). For instance, PEs originating from 1–20 $\mu$m sea salt aerosols exhibit a wide range of concentrations from $10^{-4}$ to $10^{-2}$ $\mathrm{cm^{-3}}$ (Jung et al., 2015; Jensen & Nugent, 2017), with a strong environmental and spatial dependency (Woodcock, 1953; Jung et al., 2015). Based on the "one in a million" definition of "lucky droplet" acting as PEs (e.g., Kostinski & Shaw, 2005), typical cloud droplet concentrations over the ocean and continents ($10^1$ to $10^3$ $\mathrm{cm^{-3}}$) imply PE concentrations of $10^{-5}$ to $10^{-3}$ $\mathrm{cm^{-3}}$. On the other hand, for climate-engineering practices such as cloud seeding, the concentration of seeded particles can exceed natural values, ranging from $10^{-1}$ to $10^1$ $\mathrm{cm^{-3}}$ (Kuba & Murakami, 2010). Due to this large variability, assessing the PE effect for a broad range of PE concentrations is important.
>
> **L99:** We choose a minimum $n_{\mathrm{PE}} = 1$ to investigate whether a 'one in a million' PE can accelerate droplet collision, as highlighted in previous studies on lucky droplets (Kostinski & Shaw, 2005; Dziekan & Pawlowska, 2017). Within a given reference volume, the minimum and maximum $n_{\mathrm{PE}}$ values of 1 and 1000 correspond to concentrations of approximately $2.97 \times 10^{-4}$ $\mathrm{cm^{-3}}$ and $2.97 \times 10^{-1}$ $\mathrm{cm^{-3}}$, respectively, reflecting the wide range of PE concentrations observed in nature (Khain, 2009; Jung et al., 2015).
>
> **L147:** For RM10, when $n_{\mathrm{PE}} = 3$, the PE number concentration is approximately $10^{-3}$ $\mathrm{cm^{-3}}$. In this case, even PEs larger than 40 $\mu$m are not effective in accelerating $t_{10\%}$ (Fig. 2b). However, when the PE concentration increases to a relatively high value ($n_{\mathrm{PE}} = 30$), PEs larger than 22 $\mu$m can substantially accelerate $t_{10\%}$ (Fig. 2b). Such high PE concentrations are uncommon but have been observed in certain oceanic conditions (Jung et al., 2015). In contrast, for RM14, which represents typical maritime clouds in a pristine environment with $N_0 = 87$ $\mathrm{cm^{-3}}$, the effect of PEs is reduced. PEs smaller than 33 $\mu$m are unable to accelerate $\mu_{100}$ regardless of $n_{\mathrm{PE}}$ (Fig. 2c). Moreover, $t_{10\%}$ is accelerated only when both $n_{\mathrm{PE}}$ and $r_{\mathrm{PE}}$ are very large (Fig. 2d). However, such extreme conditions

are uncommon in typical maritime environments. This suggests that the impact of PEs depends on the initial DSD shape, requiring a collisionally stable cloud for a substantial effect.

**L315:** In this study, PEs larger than 22 $\mu$m are found to effectively accelerate the precipitation ($t_{10\%}$) for clouds in relatively polluted environments, when their concentration exceeds $10^{-3}$ cm$^{-3}$, consistent with Feingold et al. (1999). While this PE concentration falls within the range observed for giant sea-salt particles over the ocean (Jung et al., 2015), for clouds in a pristine environment, substantially higher numbers and sizes of PE are required to achieve effective precipitation acceleration. These observations are based on measurements of sea-salt aerosols (2–20 $\mu$m), which have large solution masses and corresponding large equilibrium sizes. However, under atmospheric conditions, these particles might not have sufficient time to grow to their equilibrium size (Ivanova et al., 1977), potentially resulting in lower PE concentrations. On the other hand, it is possible that PE concentrations can increase through stochastic collisions (Kostinski & Shaw, 2005; Dziekan & Pawlowska, 2017) as the cloud evolves. Furthermore, because the critical threshold decreases in DSDs with higher *collisional stability*, the effect of PEs is expected to be especially strong in non-precipitating, or polluted clouds as suggested in previous studies (e.g., Johnson, 1993; Dziekan et al., 2021).

**4. Comment 4**

**RC:** *Interpretation of Function $\Phi$ in Equation (5): The function $\Phi$, defined in Equation (5), describes the impact of PEs by combining their number and size. However, it is a decreasing function of PE size and concentration, which makes the plots harder to interpret. It might be more intuitive if $\Phi$ were an increasing function, and if $\Phi$ = 0 corresponded to the absence of PEs.*

**AR:** We appreciate this reviewer's comment. We have re-defined $\Phi = n_{\mathrm{PE}}^{a} r_{\mathrm{PE}}^{b}$. Now, $\Phi = 0$ represents the absence of PEs. We believe that Equation (5) becomes more intuitive due to this change.

**L160:** Thus, we write

$$\mu_\alpha = c_\alpha - k_\alpha n_{\mathrm{PE}}^{a_\alpha} r_{\mathrm{PE}}^{b_\alpha} = c_\alpha - k_\alpha \Phi_\alpha(n_{\mathrm{PE}}, r_{\mathrm{PE}}) \qquad (1)$$

for a $\mu_\alpha$ exceeding the critical threshold.

**5. Comment 5**

**RC:** *Definition of Rain in the 10 % Mass Conversion: It is unclear how rain is defined when calculating the time required to convert 10 % of the total cloud mass to rain. Is the initial presence of rainwater (e.g., from PEs already exceeding the rain threshold at t = 0 s) included in the 10 % mass calculation?*

**AR:** It is defined as 10 % of initial $q_c + q_r$, where $q_r > 0$ when $r_{PE} > 40$ $\mu$m and $n_{PE} > 0$. Now, we explain this in the manuscript more clearly.

**L131:** Adding PEs increases the initial $q_{\mathrm{c}}$, or the rainwater mixing ratio $q_{\mathrm{r}}$ when $r_{\mathrm{PE}} > 40$ $\mu$m and $n_{\mathrm{PE}} > 0$, potentially limiting the comparability of simulated cases. To address this, we restricted the analysis of $t_{10\,\%}$ and further conversion rates such as autoconversion rate (i.e., raindrop formation by

collisions between cloud droplets), and accretion rate (i.e., raindrop growth by raindrops collecting cloud droplets) to cases where the increase in the initial $q_{\mathrm{c}}+q_{\mathrm{r}}$ due to the addition of PEs is below 2 %. In most cases, the increase in $q_{\mathrm{c}}$ and $q_{\mathrm{r}}$ is below 1 %. However, two exceptions, $n_{\mathrm{PE}}=300,\ \mathrm{with}\ r_{\mathrm{PE}}=40\,\mu\mathrm{m}$ and $n_{\mathrm{PE}}=1000,\ \mathrm{with}\ r_{\mathrm{PE}}=27\,\mu\mathrm{m}$, show an increase of 1.9 %.

**6. Comment 6**

**RC:** *Interpretation of Results Regarding the 'Lucky Droplet' Theory: The statement on line 190—"This suggests that while PEs can accelerate the formation of the largest raindrops, these droplets may not substantially impact overall rain initiation after the initial period when they are few"—is described as contradicting the 'lucky droplet' theory. However, I disagree with this interpretation. The results show that when there are too few lucky droplets (PEs), they do not significantly impact the time to convert $10\,\%$ of the cloud mass into rain. Nonetheless, when more lucky droplets are present, they do affect this time ($t10\,\%$). Furthermore, even if $t10\,\%$ averaged over small cloud volumes is not affected, the presence of lucky droplets could still play an important role in rain formation in real clouds.*

**AR:** We agree that the original statement was misleading. We have revised the argument to state that while $t_{100}$ does not necessarily ensure a shorter $t_{10\%}$ when PEs are too few, the presence and continued generation of lucky droplets can indeed increase the PE concentration and influence rain initiation.

> **L234:** This suggests that while PEs can accelerate the formation of the largest raindrop, these droplets may not directly impact the overall rain mass growth when the number of PEs is low.
>
> **L321:** On the other hand, it is possible that PE concentrations can increase through stochastic collisions (Kostinski & Shaw, 2005; Dziekan & Pawlowska, 2017) as the cloud evolves.

**7. Technical Comments**

**RC:** *Line 135: There is an unclosed parenthesis.*

**AR:** Done.


**Authors' Response to Reviews of**

**The Critical Number and Size of Precipitation Embryos to Accelerate Warm Rain Initiation**

Jung-Sub Lim, Yign Noh, Hyunho Lee, Fabian Hoffmann
*Atmospheric Chemistry and Physics,*
* * *
RC: *Reviewers' Comment*,     AR: Authors' Response,     ☐ Manuscript Text

*All line numbers in this response letter refer to the revised manuscript unless stated.*

**RC:** *This paper offers a fresh perspective on the role that rare large droplets, here named "precipitation embryos (PEs)", can have on precipitation formation from a droplet distribution which would otherwise undergo slower collision-coalescence (akin to the "luck-drops" theory). The study is well-motivated and the use of the superdroplet model throughout is particularly appropriate. The conclusions drawn are well-grounded and are worthwhile for our understanding of the timing of rain formation. They certainly further our knowledge of how sufficiently numerous and large droplets can accelerate droplet collision-coalescence, and they also give particularly useful insight to how turbulence-induced collision enhancement (TICE) - another process known to accelerate rain formation - can either complement or inhibit the acceleration caused by large droplets. Indeed I believe the study to be within the scope of ACP and I recommend that the editor consider publishing this work. Nevertheless I have some specific comments I would like to see addressed first, particularly with regard to section 3 and the decision to truncate the droplet size distribution (DSD) for almost all the analysis.*

**AR:** Dear reviewer, we highly appreciate your effort and time to evaluate this manuscript and provide fruitful comments that greatly improved the quality of the paper. We have revised our manuscript accordingly. Below you will find a point-by-point response from the authors.

**1. Comment 1**

**RC:** *Most critically, I am unconvinced by the decision to truncate the DSD at 20 m and I believe the analysis and conclusions of the paper would be clearer if they were presented without this cut-off. The truncation is justified as creating a "stable initial DSD in which collisions are negligible" however, as for example the black lines in figure 6 show, the truncated DSDs do undergo frequent collisions and so the truncation appears unjustified. Contrary to the Author's introduction, I believe such examples also mean the role of "precipitation embryos" (PEs) in inducing collisions in an otherwise stable distribution is not addressed in this paper. The paper is better introduced as showing how PEs can accelerate the timing of rain formation for a DSD in which droplets already can collide. I am of the opinion that the analysis would be clearer, and the conclusions more convincing, if the cut-off was removed. For example the red and green lines in figures 4(a) and 4(b) (i.e. the RM12 and RM14 setups) show large changes between figures 4(a) and 4(b), and 4(c) and 4(d), ie. when the un-truncated distribution is used instead. Likewise figure 5 shows large differences in the results without truncation. This means the behaviour of the critical threshold is heavily influenced by the cutoff at 20 m and the analysis of the role of PEs, particularly in section 3, would be more convincingly shown if that influence wasn't present.*

**AR:** We also agree with the reviewer that DSD without cut-off represents a more natural DSD. The original

motivation for the cut-off was to isolate the effects of PEs by starting from an initially *stable* DSD, in which droplets are not large enough to initiate frequent collisions. However, as the reviewer correctly pointed out, collisions can still occur in the truncated DSDs, even without PEs. To address this, we have revised the manuscript by making the case without the DSD cut-off (*broad cases*) as the base case and renamed the cases with DSD cut-off as *narrow* cases.

AR: In particular, Sec. 3 now shows the results without the DSD cut-off. The *narrow* cases are only used to discuss how the PE critical value changes with DSD shape, particularly with respect to the largest initial droplet size. We note that in cases where $\bar{r} = 10 \ \mu$m (light blue lines in Fig.1), there are very few cloud droplets larger than $20\mu$m even without the DSD cut-off. Thus, the conclusions in Sec. 3 are not substantially different from the original manuscript.

**2. Comment 2**

RC: *L3 (and repeated): please reconsider the use of the term "colloidally stable". The terminology is misleading because it is conventionally used to discuss the condition of a colloid substance - i.e. the mixture of soluble and insoluble substances in a solute.*

AR: Although Squires (1958) originally used the term "colloidal stability" to describe the ability of a DSD to initiate precipitation, we agree with the reviewer that this term has a different conventional meaning in other areas of the scientific community. Therefore, we decided to use the term *collisional stability*, and defined it in the introduction.

> **L16:** These collision-limited DSDs can be regarded as being in a *collisionally stable* state (Squires, 1958), where mechanisms that accelerate the collision-coalescence process to form raindrops and initiate precipitation are crucial for breaking this stability.

**3. Comment 3**

RC: *L22 ("so-called precipitation embryos (PEs)"): is the term "precipitation embryo" coined in this paper? If not, please provide a citation here. If so, please consider if there is any existing terminology you can use instead. Why for example is the term "lucky droplet" not appropriate?*

AR: The term precipitation embryos was originally used to describe large droplets that initiate warm rain in earlier studies (e.g., Pontikis et al., 1987; Johnson, 1993). In this study, we focus on the effects of large droplets that initiate warm rain, regardless of their origin—whether from giant aerosol particles, "lucky droplets" arising from stochastic collisions, or seeded particles. While the term "lucky droplet" implies a droplet form from a series of unlike stochastic collisions, and "giant aerosol" refers to initially large droplets, we find that the broader concept of precipitation embryos which encompasses large droplets formed through multiple potential mechanisms, most suitable for this study. We have cited Johnson (1993), as it provides a definition most closely aligned with the concept used in our research.

> **L24:** ... and (iii) (iii) the role of so-called precipitation embryos (PEs) (e.g., Johnson, 1993), the primary focus of this study.

**4. Comment 4**

**RC:** *L81 ("$N_0$ = 238 cm-3, 456 cm-3 and 523 cm-3"): the values for $N_0$ given here do not match those in the legend of figure 1 so please correct them. Also it would be clearer if their order was consistent with the ordering of in the sentence before (i.e. if $\bar{r}$ is ascending, $N_0$ is descending).*

**AR:** Thank you for pointing this out. We have corrected the values in the text to match those in the legend of Figure 1, which were correct. We also reordered the values to ensure that when $\bar{r}$ is ascending, $N_0$ is descending, as suggested.

> **L92:** In these cases, $N_0 = 466,\ 138,$ and $87\ \mathrm{cm}^{-3}$ to achieve the same $q_c = 1.0\ \mathrm{g\ kg}^{-1}$ (Fig. 1).

**5. Comment 5**

**RC:** *Figure 1: These DSDs do not show very clearly the alteration you make by truncating these distributions, nor do they show the impact of adding PEs on them. It would be helpful to the reader to include a neighbouring figure (or to replace this one) with a plot that shows the truncated DSDs and some DSDs which example the distribution with $n_{\mathrm{PE}}, r_{\mathrm{PE}}$ values.*

**AR:** Thank you for the recommendation. We have incorporated a second panel in Figure 1 to address your suggestion. This panel now illustrates the truncated DSDs and includes an example PE distribution with $r_{\mathrm{PE}} = 27\ \mu\mathrm{m}$ and different $N_{\mathrm{PE}}$ values to provide a comprehensive representation of the DSD cut-off and PE addition. Moreover, the DSD with added PE ($r_{\mathrm{PE}} = 27\ \mu\mathrm{m}$ and $n_{\mathrm{PE}} = 1000$) for RM10 is shown as the dashed line in panel (a).

[Figure]

Figure 1: (a) Initial DSDs for various $\bar{r}$ and their corresponding $N_0$ values. The dashed line represents the DSD with $r_{\mathrm{PE}} = 27\ \mu\mathrm{m}$ and $n_{\mathrm{PE}} = 1000$. b) Initial DSDs with a DSD cut-off for various $\bar{r}$ and $N_0$ values, along with a vertical bar plot showing various PE distributions for $r_{\mathrm{PE}} = 27\ \mu\mathrm{m}$ and various $n_{\mathrm{PE}}$ values. **Excerpted from Fig. 1 of the manuscript.**

**6. Comment 6**

**RC:** *L96 ("To explore the impact of PEs, we investigate 49 ensemble simulations…"): it's quite important that this is justified with a comparison to the real world. How do these PE radii and number concentrations compare with observations? Especially relevant would be to put them in the context of shallow convective clouds, since that's the range of TICE you consider later on, or cloud seeding. Context is particularly necessary because figure 2 shows both $\mu_{100}$ and $\mu_{10\%}$ have almost no dependency on a large portion of the $n_{\mathrm{PE}}, r_{\mathrm{PE}}$ space spanned by the ensemble.*

**AR:** We agree with the reviewer that providing the observed range of $n_{\mathrm{PE}}$ and $r_{\mathrm{PE}}$ values is essential for context. In the revised manuscript, we have included references to observed PE number concentrations across different cloud types. Notably, the smallest $n_{\mathrm{PE}} = 1$ is lower than typically observed PE concentrations. However, this value was included to explore the "lucky droplet" effect, where a single "one-in-a-million" droplet is expected to accelerate the collision process. Now we relate the PE concentrations considered in this study to the observed values over the ocean to incorporate the reviewer's comment.

**AR:** In addition, we have excluded cases with $r_{\mathrm{PE}} = 15\ \mu$m in the revised manuscript to improve clarity. While these cases were initially included to demonstrate the limited effect of small PEs ($r_{\mathrm{PE}} < 20\ \mu$m), the effect is sufficiently represented by the cases with $r_{\mathrm{PE}} = 18\ \mu$m making the discussion clearer.

> **L45:** Lastly, there is a large uncertainty in the number concentration of PEs in clouds (Khain, 2009). For instance, PEs originating from 1–20 $\mu$m sea salt aerosols exhibit a wide range of concentrations from $10^{-4}$ to $10^{-2}$ cm$^{-3}$ (Jung et al., 2015; Jensen & Nugent, 2017), with a strong environmental and spatial dependency (Woodcock, 1953; Jung et al., 2015). Based on the "one in a million" definition of "lucky droplet" acting as PEs (e.g., Kostinski & Shaw, 2005), typical cloud droplet concentrations over the ocean and continents ($10^1$ to $10^3$ cm$^{-3}$) imply PE concentrations of $10^{-5}$ to $10^{-3}$ cm$^{-3}$. On the other hand, for climate-engineering practices such as cloud seeding, the concentration of seeded particles can exceed natural values, ranging from $10^{-1}$ to $10^1$ cm$^{-3}$ (Kuba & Murakami, 2010). Due to this large variability, assessing the PE effect for a broad range of PE concentrations is important.
>
> **L99:** We choose a minimum $n_{\mathrm{PE}} = 1$ to investigate whether a 'one in a million' PE can accelerate droplet collision, as highlighted in previous studies on lucky droplets (Kostinski & Shaw, 2005; Dziekan & Pawlowska, 2017). Within a given reference volume, the minimum and maximum $n_{\mathrm{PE}}$ values of 1 and 1000 correspond to concentrations of approximately $2.97 \times 10^{-4}$ cm$^{-3}$ and $2.97 \times 10^{-1}$ cm$^{-3}$, respectively, reflecting the wide range of PE concentrations observed in nature (Khain, 2009; Jung et al., 2015).
>
> **L147:** For RM10, when $n_{\mathrm{PE}} = 3$, the PE number concentration is approximately $10^{-3}$ cm$^{-3}$. In this case, even PEs larger than 40 $\mu$m are not effective in accelerating $t_{10\%}$ (Fig. **??**b). However, when the PE concentration increases to a relatively high value ($n_{\mathrm{PE}} = 30$), PEs larger than 22 $\mu$m can substantially accelerate $t_{10\%}$ (Fig. **??**b). Such high PE concentrations are uncommon but have been observed in certain oceanic conditions (Jung et al., 2015). In contrast, for RM14, which represents typical maritime clouds in a pristine environment with $N_0 = 87$ cm$^{-3}$, the effect of PEs is reduced. PEs smaller than 33 $\mu$m are unable to accelerate $\mu_{100}$ regardless of $n_{\mathrm{PE}}$ (Fig. **??**c). Moreover, $t_{10\%}$ is accelerated only when both $n_{\mathrm{PE}}$ and $r_{\mathrm{PE}}$ are very large (Fig. **??**d). However, such extreme conditions are uncommon in typical maritime environments. This suggests that the impact of PEs depends on the initial DSD shape, requiring a collisionally stable cloud for a substantial effect.

**L315:** In this study, PEs larger than $22\ \mu$m are found to effectively accelerate the precipitation ($t_{10\%}$) for clouds in relatively polluted environments, when their concentration exceeds $10^{-3}\ \mathrm{cm}^{-3}$, consistent with Feingold et al. (1999). While this PE concentration falls within the range observed for giant sea-salt particles over the ocean (Jung et al., 2015), for clouds in a pristine environment, substantially higher numbers and sizes of PE are required to achieve effective precipitation acceleration. These observations are based on measurements of sea-salt aerosols ($2$–$20\ \mu$m), which have large solution masses and corresponding large equilibrium sizes. However, under atmospheric conditions, these particles might not have sufficient time to grow to their equilibrium size (Ivanova et al., 1977), potentially resulting in lower PE concentrations. On the other hand, it is possible that PE concentrations can increase through stochastic collisions (Kostinski & Shaw, 2005; Dziekan & Pawlowska, 2017) as the cloud evolves. Furthermore, because the critical threshold decreases in DSDs with higher *collisional stability*, the effect of PEs is expected to be especially strong in non-precipitating, or polluted clouds as suggested in previous studies (e.g., Johnson, 1993; Dziekan et al., 2021).

**7. Comment 7**

**RC:** *L102 ("larger PEs can substantially increase the initial $q_c$"): please state how much your simulations are perturbed by the presence of PEs. For example stating what the largest $q_c$ of your ensemble is (when $n_{\mathrm{PE}}, r_{\mathrm{PE}}$ = 1000,18 )*

**AR:** Thank you for this comment. In response, we have quantified the perturbations caused by the addition of PEs. The maximum increases in $q_c + q_r$ are $12.5\%$ and $6.4\%$, respectively, which were excluded from the original analysis. To address this, we have now introduced a criterion whereby only cases are included in the analysis in which $q_c + q_r$ increase less than $2\%$ due to the addition of PEs (see the values in Fig. 2 of this letter). For most cases, the increase in $q_c + q_r$ caused by PEs is below $1\%$.

**L131:** Adding PEs increases the initial $q_{\mathrm{c}}$, or the rainwater mixing ratio $q_{\mathrm{r}}$ when $r_{\mathrm{PE}} > 40\ \mu$m and $n_{\mathrm{PE}} > 0$, potentially limiting the comparability of simulated cases. To address this, we restricted the analysis of $t_{10\ \%}$ and further conversion rates such as autoconversion rate (i.e., raindrop formation by collisions between cloud droplets), and accretion rate (i.e., raindrop growth by raindrops collecting cloud droplets) to cases where the increase in the initial $q_{\mathrm{c}} + q_{\mathrm{r}}$ due to the addition of PEs is below 2 %. In most cases, the increase in $q_{\mathrm{c}}$ and $q_{\mathrm{r}}$ is below 1 %. However, two exceptions, $n_{\mathrm{PE}} = 300,\ \mathrm{with}\ r_{\mathrm{PE}} = 40\ \mu$m and $n_{\mathrm{PE}} = 1000,\ \mathrm{with}\ r_{\mathrm{PE}} = 27\ \mu$m, show an increase of 1.9 %.

**8. Comment 8**

**RC:** *L213 ("although $t_{100}$ can be shorter than in the cases without PEs."): please provide a suggested explanation or citation for this statement.*

**AR:** This sentence refers that $t_{100}$ is shorter in the cases with small $n_{\mathrm{PE}}$ values compared to the $t_{100}$ value in the case without PEs. We acknowledge the reviewer's concern about the ambiguity and changed the sentence accordingly.

[Figure]

Figure 2: Heatmap of change in initial $q_c + q_r$ by PEs for different $r_{\mathrm{PE}}$ and $n_{\mathrm{PE}}$ values.

> **L253:** Thus, although larger PEs are more likely to collide, the overall collision frequency remains low when $n_{\mathrm{PE}}$ is small, resulting in slower PE autoconversion compared to non-PE autoconversion. While non-PE-autoconversion always decreases with increasing $n_{\mathrm{PE}}$, PE-autoconversion increases substantially only for $n_{\mathrm{PE}} \geq 100$. Therefore, before exceeding the critical threshold, PEs suppress non-PE autoconversion more than they enhance autoconversion which can even lead to a decrease in the total (PE and non-PE) autoconversion. Hence, shorter $t_{100}$ does not necessarily lead to a shorter $t_{10\%}$ when $n_{\mathrm{PE}}$ is small (Fig. 2).

**9. Comment 9**

**RC:** *Figure 6: With increasing PE concentration, the accretion and auto-conversion rates due to non-PE collisions maximise earlier. A acknowledgement and a possible reason for this behaviour would be beneficial because it is not self-evident why it should occur.*

**AR:** This behavior occurs because PE particles suppress non-PE autoconversion, as discussed in this paper. As shown in Fig. 6c, the non-PE autoconversion rate is nearly identical across all cases during the initial 1000–1200 s, regardless of PE concentration. However, when the PE effect is stronger, more non-PE droplets are collected by PEs, reducing the number of droplets available for autoconversion. As a result, in cases with higher PE concentrations, the autoconversion rate peaks earlier and then begins to decline. This suppression of non-PE autoconversion also decreases the number of non-PE raindrops, which reduces the non-PE accretion rate and causes it to peak earlier in these cases. These findings suggest that the primary role of PEs is to collect non-PE droplets, while indirectly suppressing both non-PE autoconversion and accretion. We have included this explanation in the revised manuscript.

> **L272:** Interestingly, at high $n_{\mathrm{PE}}$, the non-PE autoconversion and accretion rates reach their peak values earlier than in cases without PEs or with low $n_{\mathrm{PE}}$ (Fig. 6c, d, e, and f). During the initial 1000 s, the non-PE autoconversion rate is nearly identical across all cases, regardless of $n_{\mathrm{PE}}$. However, when $n_{\mathrm{PE}}$ is high, more non-PE droplets are collected by PEs, reducing the number of droplets available for autoconversion. As a result, the non-PE autoconversion rate peaks and declines earlier in cases with higher PE concentrations. This suppression of non-PE autoconversion decreases the number of non-PE raindrops and the non-PE accretion rate. These findings highlight that the primary role of PEs is to collect non-PE droplets, which might suppress non-PE autoconversion and accretion.

**10. Technical Comments**

**RC:** *L11 ("A key question in warm rain initiation is to explain..."): There is no question posed here, consider rephrasing the second half of the sentence into a question, or rewording "a key question".*

AR: Done. Now, we address this as a key challenge.

**RC:** *L36 ("In particular, if we consider ... Thus, it is also important to account for stochastic fluctuations in the collision process."): please rephrase this sentence because this logic is hard to follow. I was left asking myself: why is a 12.5 $\mu m$ droplet relevant to your study, and how does this citation justify the need for stochastic fluctuations? I assume you mean than by including stochasticity you don't need as large lucky droplets to initiate collisions, but please make this more explicit if so.*

AR: We agree with the reviewer that the reference to a 12.5 $\mu m$ droplet is not directly relevant to our study. The original intention was to highlight that including stochasticity reduces the need for larger or more PEs to initiate rain. To address the reviewer's concerns, we have rephrased the sentence to clarify this point and provide a better context for the importance of collision stochasticity.

> **L39:** In the absence of PEs, DSDs with small-sized droplets barely initiate precipitation unless stochastic fluctuations in the collision process are considered. This phenomenon is known as the "lucky droplet" effect, which may produce PEs on its own (Telford, 1955; Kostinski & Shaw, 2005; Dziekan & Pawlowska, 2017). When this effect dominates, adding only a few PEs may not substantially accelerate rain initiation.

**RC:** *L56 ("frequently called superdroplets by introducing a weighting factor"): please add missing comma after "superdroplets".*

AR: Done

**RC:** *L110 ("The timescale $t_{10\,\%}$ represents the time when $10\,\%$ of the initial cloud droplet mass converts to rain..."): is rain here defined as above 40 or 100 microns?*

AR: A raindrop is defined as a droplet larger than 40 $\mu$m, in radius in our study. We clarified this in the revised manuscript.

**RC:** *L145 (equation 6): Missing subscript on $\alpha$ on $\Phi$ the left-hand side of the equation.*

AR: Done.

**RC:** *Figure 4 and figure 5: Please reconsider the colours you use to plot these figures because they are unkind*

*to colour-blindness (especially figure 4). Also in figure 4, the re-use of the same colours for subplots (e) and (f), although they are no longer denoting different $\bar{r}$, is undesirable.*

AR: Thank you for your comment. We have now revised all figures to be accessible for color vision deficiency. Also, we assigned new colors to cases with the same $\bar{r}$ but different $\varepsilon$ in panels e and f of Figure 4, ensuring that these colors do not overlap with those used in panels a–d.

[revised manuscript text omitted]